# Learning Robust Spectral Dynamics for Temporal Domain Generalization

**En Yu, Jie Lu,**\* **Xiaoyu Yang, Guangquan Zhang, Zhen Fang**

Australian Artificial Intelligence Institute (AAII),
University of Technology Sydney, Australia.
{en.yu-1,jie.lu,guangquan.zhang,zhen.fang}@uts.edu.au;
Xiaoyu.Yang-3@student.uts.edu.au

## Abstract

Modern machine learning models struggle to maintain performance in dynamic environments where temporal distribution shifts, *i.e., concept drift*, are prevalent. Temporal Domain Generalization (TDG) seeks to enable model generalization across evolving domains, yet existing approaches typically assume smooth incremental changes, struggling with complex real-world drifts involving both long-term structure (incremental evolution/periodicity) and local uncertainties. To overcome these limitations, we introduce FreKoo, which tackles these challenges through a novel frequency-domain analysis of parameter trajectories. It leverages the Fourier transform to disentangle parameter evolution into distinct spectral bands. Specifically, the low-frequency components with dominant dynamics are learned and extrapolated using the Koopman operator, robustly capturing diverse drift patterns including both incremental and periodic drifts. Simultaneously, potentially disruptive high-frequency variations are smoothed via targeted temporal regularization, preventing overfitting to transient noise and domain uncertainties. In addition, this dual-spectral strategy is rigorously grounded through theoretical analysis, providing stability guarantees for the Koopman prediction, a principled Bayesian justification for the high-frequency regularization, and culminating in a multiscale generalization bound connecting spectral dynamics to improved generalization. Extensive experiments demonstrate FreKoo's significant superiority over state-of-the-art TDG methods, particularly excelling in real-world streaming scenarios with complex drifts and uncertainties.

## 1 Introduction

Modern machine learning models face significant challenges in dynamic environments where data distributions evolve over time [1]. Unlike static settings that assume an IID relationship between training and test data, real-world scenarios often involve continuously generated data streams (e.g., user activity logs, sensor readings, financial transactions) exhibiting temporal dependencies and non-stationarity due to factors like shifting user behavior, environmental variations, or system changes [2]. These temporal dynamics lead to distributional shifts over time, known as *concept drift*, which invalidates the alignment between historical and future out-of-distribution (OOD) data [3–5]. Consequently, models trained on past data frequently fail to generalize to future instances, leading to performance degradation and reduced reliability. This has spurred increasing interest in *Temporal Domain Generalization (TDG)*, which seeks to develop models capable of generalizing robustly across chronologically ordered source domains to unseen future target distributions [6].

---

\*Correspondence to Jie Lu and Zhen Fang

39th Conference on Neural Information Processing Systems (NeurIPS 2025).

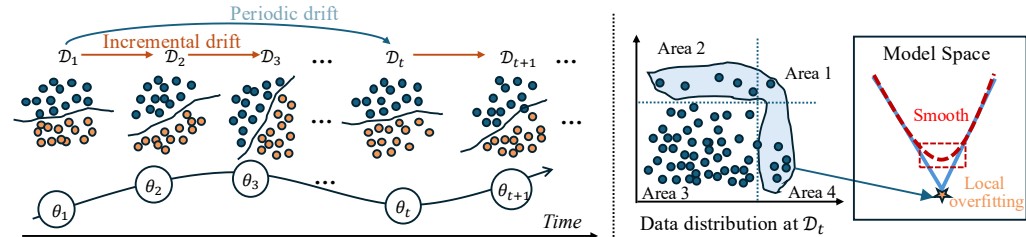

Figure 1: Illustration of challenges in TDG. Left: Complex drifting situations can involve both incremental shifts (e.g., $\mathcal{D}_1 \to \mathcal{D}_2 \to \mathcal{D}_3$) and long-term periodic returns (e.g., $\mathcal{D}_t$ resembling $\mathcal{D}_1$ after a cycle). The underlying optimal parameters $\theta_t$ evolve accordingly. Right: Within any domain $\mathcal{D}_t$, uncertainties or non-IID data concentrated in Areas 1, 2 and 4 compared to data in Area 3 can lead to local overfitting (solid blue line). Robust generalization requires converging to a smoother area (red dashed line) that is less sensitive to such localized noise or outliers.

Recent progress in TDG falls into two categories: *data-driven* [7] and *model-centric* [8] approaches. *Data-driven methods* enhance temporal robustness by simulating future distributions via OOD data generation [9, 10] or learning temporally invariant features [11]. *Model-centric methods* often leverage dynamic modeling, such as time-sensitive regularization or parameter forecasting, to adapt models to evolving distributions [6, 12]. However, existing methods often implicitly assume that concept drift is primarily incremental or focus on ensuring local smoothness between adjacent domains. This overlooks a crucial aspect of many real-world scenarios: *1) Difficulty in Modeling Long-term Periodic Dynamics*: real-world concept drift frequently exhibits long-term periodicity (e.g., seasonality, weekly/daily user patterns, economic cycles), not just smoothly incremental changes [13, 14], as shown in the left of Figure 1. Current approaches struggle to capture these recurring patterns spanning longer time horizons [8]. Their effectiveness often diminishes, particularly near phase transitions of periodic cycles where the inability to anticipate pattern recurrence hinders generalization. *2) Vulnerability to Overfitting Domain-Specific Uncertainties*: current TDG methods typically segment continuous data streams into discrete temporal domains. However, the complex drift patterns inherent in real-world streams make it difficult to guarantee that data within each domain remains IID, as visualized in Appendix D.1 Figure 5. Consequently, localized uncertainties and non-IID characteristics within domains can complicate optimization and increase the risk of overfitting to domain-specific artifacts (illustrated in the right of Figure 1), ultimately undermining stable cross-temporal generalization [15].

These challenges underscore the necessity for principled frameworks capable of capturing long-term incremental/periodic dynamics while filtering local uncertainties. Intuitively, the trajectory of model parameters over time provides a comprehensive reflection of the underlying concept drift, encapsulating its long-range evolution alongside its unpredictable uncertainties. Thus, we turn to frequency-domain analysis of the parameter trajectories. Frequency-domain analysis inherently isolates dominant dynamics by representing them as distinct spectral peaks, facilitating robust modeling of long-term evolving patterns [16]. Moreover, as analyzed in [17], spectral representations can reveal dynamic temporal structures often obscured by conventional statistical measures (e.g., mean, variance), thereby providing a richer understanding of parameter evolution. Specifically, transforming trajectories into the frequency domain reveals that dominant evolving trends and periodic components typically manifest as energy concentrations at low frequencies, whereas transient uncertainties typically dominate higher frequencies [18, 19]. This spectral separation naturally allows for targeted dynamic modeling: isolating and modeling the predictable long-term low-frequency dynamics while permitting robust management of disruptive high-frequency variations. Compared to purely time-domain approaches, this frequency-domain perspective offers a more robust mechanism for modeling diverse temporal dynamics in real-world scenarios with complex drifts and uncertainties.

Motivated by this, we propose **FreKoo**, a *Frequency-Koopman Regularized* framework that enhances temporal generalization by integrating spectral decomposition with Koopman operator theory. Specifically, as shown in Figure 2, FreKoo decomposes model parameter trajectories into low-frequency and high-frequency components via the Fourier transform. Koopman operator theory [20, 21] is employed to learn a stable linear model for the evolution of low-frequency dynamics, enabling robust prediction of incremental trends and periodicity. Simultaneously, it introduces a targeted temporal smooth regularization to the high-frequency components, promoting stable convergence and preventing overfitting to local uncertainties. Furthermore, we derive a multiscale generalization

bound that characterizes the interplay between Koopman stability, temporal regularization, and generalization robustness. This theoretical foundation provides rigorous insight into how FreKoo improves generalizability in dynamic environments, offering a unified and principled solution to complex concept drifts in TDG. Our main contributions are summarized as follows:

- We pioneer a spectral analysis perspective on parameter trajectories for TDG. This enables principled disentanglement of complex and multiscaled temporal dynamics, offering a novel pathway to address limitations of prior methods in handling long-term periodicity and domain-specific uncertainties.

- We propose FreKoo, a novel end-to-end framework that materializes this spectral insight. It synergistically combines Koopman operator extrapolation for stable low-frequency dynamics with targeted regularization of high-frequency uncertainties, thereby enhancing robustness against complex drifting situations.

- We establish a rigorous theoretical foundation for FreKoo via a novel multiscale generalization bound. It connects FreKoo's spectral dominant dynamics and temporal smooth regularization to stable generalization. Extensive experiments further demonstrate FreKoo's significant superiority particularly in real-world scenarios with periodicity and uncertainties.

## 2  Related Works

**Temporal Domain Generalization (TDG).** TDG specifically tackles scenarios where data distributions evolve over time (i.e., concept drift), aiming to train models on historical data that generalize to near future domains [6]. This requires explicitly capturing and leveraging the dynamics of the distribution shifts over time. Current TDG research broadly follows two main directions. Data-driven approaches seek temporal robustness by either synthesizing future data via OOD generation [7, 9, 10], or learning time-invariant representations [11]. Model-centric methods focus on incorporating dynamic adaptation mechanisms directly into the learning process, such as time-sensitive regularizations that encourage smooth decision boundaries over time [6] or forecasting future parameters based on past evolution [8]. Furthermore, recognizing the continuous nature of time, Continuous TDG (CTDG) methods have been proposed, often treating time as a continuous index for adaptation [12, 22]. However, despite progress, effectively modeling long-term periodic patterns and mitigating sensitivity to intra-domain uncertainties leading to local overfitting remain key challenges across existing TDG methods.

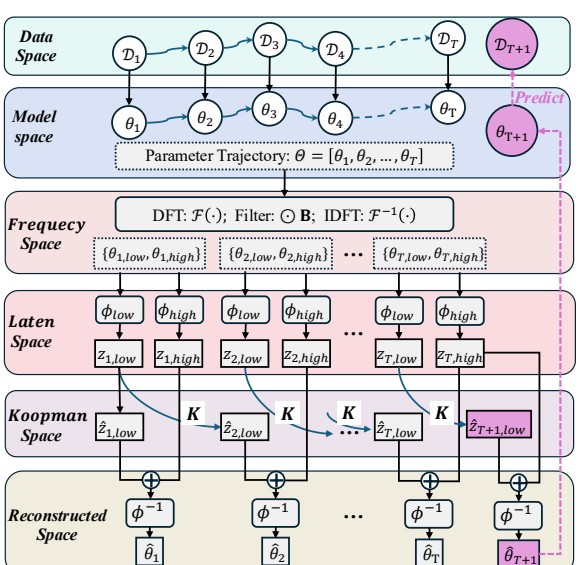

Figure 2: FreKoo Framework. It decomposes model parameter trajectories into low-frequency and high-frequency components via the Fourier transform. Then, the Koopman operator is employed to learn the evolution of low-frequency dynamics, enabling robust prediction of incremental trend and periodicity. Also, it introduces a targeted temporal difference regularization to the high-frequency components, promoting smooth convergence and preventing overfitting to local uncertainties.

**Concept Drift.** Concept drift signifies a change in the underlying data distribution over time, i.e., $P_{t+1}(X, y) \neq P_t(X, y)$, fundamentally challenges model reliability by requiring dynamic adaptation [3]. Concept drift manifests in various forms, including sudden, gradual, incremental, and recurring (periodic) patterns. Much prior research treats it as unpredictable and often employs detect-then-adapt strategies [23–25]. In contrast, TDG aims to leverage continuous or predictable evolutionary dynamics for proactive generalization [8]. However, existing TDG methods primarily model smooth incremental changes, consequently struggling to capture complex, long-range structures

like periodicity which are common in real-world streams and demand modeling approaches beyond simple monotonic progression assumptions.

**Frequency Learning.** Frequency learning has emerged as a promising paradigm for modeling non-stationary temporal dynamics in machine learning [26, 27]. By transforming time-domain signals into the spectral space, frequency-aware methods effectively capture periodic patterns and multi-scale trends that remain obscured in raw signals. For instance, prior works [28–30] leverage spectral decomposition to identify periodic structures in temporal data. FAN [19] further demonstrates that frequency-domain representations can reveal dynamic features—including long-term trends and seasonal variations—that traditional statistical measures (e.g., mean, variance) fail to characterize. These insights provide a foundational basis for our work, as we extend frequency learning to a novel perspective, modeling parameter trajectories under concept drift.

## 3 Methodology

### 3.1 Preliminary

**Temporal Domain Generalization.** Given a sequence of $T$ temporal source domains $\{\mathcal{D}_t\}_{t=1}^T$, each domain at timestamp $t$ is defined as $\mathcal{D}_t = \{(\mathbf{x}_t^{(i)}, y_t^{(i)})\}_{i=1}^{N_t}$, where $(\mathbf{x}_t^{(i)}, y_t^{(i)}) \in \mathcal{X}_t \times \mathcal{Y}_t$ is a sample-label pair, $N_t$ is the number of samples of $\mathcal{D}_t$, and $\mathcal{X}_t$ and $\mathcal{Y}_t$ denote the input and label spaces at timestamp $t$. We assume that each source domain $\mathcal{D}_t$ is drawn from a time-dependent distribution $P_t(X, Y)$, and the sequence $\{P_t\}_{t=1}^T$ exhibits *concept drift*, which may include incremental or long-term periodic trends. The objective of TDG is to build a model $g(\cdot; \theta_t) : \mathcal{X}_t \to \mathcal{Y}_t$ using historical source domains $\{\mathcal{D}_t\}_{t=1}^T$ that generalizes effectively to an OOD future target domain $\mathcal{D}_{T+1} \sim P_{T+1}(X, Y)$, without access to $\mathcal{D}_{T+1}$ during training [6].

**Challenges.** Prevailing TDG methods, typically constrained by assumptions of incremental change or local smoothness, struggle to handle complex concept drifts. This manifests in two primary challenges. First, they fail to capture long-range periodic dynamics (*Challenge 1*), where the data distribution periodically recurs after $L$ steps, i.e., $\exists, L \in \mathbb{Z}^+$ s.t. $P_t(X, Y) \approx P_{t+kL}(X, Y)$ for all $k \in \mathbb{Z}^+$, despite undergoing short-term shifts $P_t(X, Y) \neq P_{t+1}(X, Y)$. Second, they are ill-equipped to filter out transient noise and domain-specific uncertainties (*Challenge 2*), a phenomenon we visualize in Appendix D.1 Figure 5. Ultimately, these limitations prevent them from effectively balancing the stability and adaptability required in real-world scenarios.

### 3.2 Proposed Method: FreKoo

To address these challenges, we model the evolution of parameters $\{\theta_1, \theta_2, \ldots, \theta_T\}$ and extrapolate to predict $\theta_{T+1}$, thereby enabling robust inference on future unseen domain $\mathcal{D}_{T+1}$ with $g(\cdot; \theta_{T+1}) : \mathcal{X}_{T+1} \to \mathcal{Y}_{T+1}$. We establish a dynamic system perspective for Temporal Domain Generalization (TDG) by modeling parameter evolution under complex drifting situations [31]. Let $\theta_t \in \mathbb{R}^D$ denote the parameter of the model $g_t : \mathcal{X}_t \to \mathcal{Y}_t$ at time $t$, the evolution over time can be described by a potentially nonlinear stochastic difference form:

$$\theta_{t+1} = \mathbf{\Phi}(\theta_t) + \epsilon_t, \tag{1}$$

where $\mathbf{\Phi} : \mathbb{R}^D \to \mathbb{R}^D$ captures deterministic transitions induced by concept drift, and $\epsilon_t$ represents stochastic perturbations. This formulation explicitly addresses concept drift by treating TDG as a parameter trajectory prediction problem governed by a stochastic nonlinear dynamical system.

Directly modeling the complex dynamics of the parameter trajectory is difficult, as it intertwines potentially different drifts and uncertainties in the real world. To overcome this, we introduce a *frequency-aware perspective* as a powerful inductive bias. We hypothesize that different frequency components within the parameter trajectory $\Theta$ correspond to distinct aspects of the concept drift: the low-frequency component captures the slowly-varying trends and the long-term periodic patterns (*Challenge 1*). The high-frequency component primarily reflects transient dynamics and domain-specific spurious noise (*Challenge 2*). By disentangling these frequency regimes, we can robustly model and extrapolate the stable structured low-frequency dynamics while suppressing the volatile, potentially misleading high-frequency variations. This insight motivates the core of our proposed FreKoo framework.

### 3.2.1 Spectral Decomposition

Given a time-varying parameter trajectory $\Theta = [\theta_1, \theta_2, \ldots, \theta_T]^\top \in \mathbb{R}^{T \times D}$ with $T$ timesteps and $D$ parameter dimensions, we utilize Fourier analysis to disentangle dominant dynamics from high-frequency fluctuations [16, 32]. Specifically, we first transfer it to frequency space via Discrete Fourier Transform (DFT) along the temporal axis, yielding a spectral representation $\Theta^f$ via:

$$\Theta^f = \mathcal{F}(\Theta), \quad \Theta^f[f, d] = \sum_{t=1}^{T} \Theta[t, d] \cdot e^{-j2\pi ft/T}, \tag{2}$$

where $\mathcal{F} : \mathbb{R}^{T \times D} \to \mathbb{C}^{N_{freq} \times D}$ denotes the DFT operator and $N_{freq} = \lfloor T/2 \rfloor + 1$ represents the number of frequency components. $f$ serves as the frequency bin index while $d$ indicates the parameter dimension index.

To identify dominant frequencies, we compute the average spectral magnitude across parameter dimensions as an energy proxy:

$$M_f = \frac{1}{D} \sum_{d=1}^{D} |\hat{\Theta}[f, d]|, \quad \forall f \in \{0, \ldots, N_{freq} - 1\}, \tag{3}$$

where $M_f \in \mathbb{R}_+$ represents the normalized energy contribution of frequency $f$. The magnitude vector $M = [M_0, M_1, \ldots, M_{N_{freq}-1}] \in \mathbb{R}_+^{N_{freq}}$ aggregates these values across all frequencies, serving as a dimension-agnostic measure of frequency importance. This averaging operation inherently suppresses parameter-wise variations in oscillation patterns, ensuring robustness to localized noise in individual dimensions. For a specified energy preservation ratio $\tau \in [0, 1]$, we select the top-$Q$ frequency indices $\mathcal{Q}_{top} = \{f_1, f_2, \ldots, f_Q\}$ where $Q = \lceil \tau N_{freq} \rceil$ and $M_{f_1} \geq M_{f_2} \geq \cdots \geq M_{f_{N_{freq}}}$, ensuring $\mathcal{Q}_{top}$ contains the $Q$ largest magnitudes. Then, we construct a frequency-domain binary mask $\mathbf{B} \in \{0, 1\}^{N_{freq} \times D}$ that preserves selected frequencies:

$$\mathbf{B}[f, d] = \begin{cases} 1 & \text{if } f \in \mathcal{Q}_{top}, \\ 0 & \text{otherwise,} \end{cases} \quad \text{for all } d = 1, \ldots, D. \tag{4}$$

Finally, $\Theta$ can be decomposed into a component $\Theta_{low}$ (dominant frequencies) and a component $\Theta_{high}$ (high-frequency fluctuations) as follows:

$$\begin{aligned} \Theta_{low}^f &= \Theta^f \odot \mathbf{B}, \quad \Theta_{high}^f = \Theta^f \odot (1 - \mathbf{B}), \\ \Theta_{low} &= \mathcal{F}^{-1}(\Theta_{low}^f), \quad \Theta_{high} = \mathcal{F}^{-1}(\Theta_{high}^f), \end{aligned} \tag{5}$$

where $\odot$ denotes element-wise multiplication, 1 represents a matrix of ones, and $\mathcal{F}^{-1}$ is the corresponding inverse DFT. By the linearity of $\mathcal{F}^{-1}$, it holds that $\Theta = \Theta_{low} + \Theta_{high}$. This decomposition enables separate analysis of dominant dynamics $\Theta_{low}$ and high-frequency fluctuations $\Theta_{high}$.

Therefore, the spectral decomposition enables targeted handling of distinct components, and Eq. (1) can be further expressed as: $\mathbf{\Phi}(\theta_t) = \mathbf{\Phi}_{low}(\theta_{t,low}) + \mathbf{R}_{high}(\theta_{t,high})$, where $\mathbf{\Phi}_{low}$ and $\mathbf{R}_{high}$ govern distinct spectral regimes. This decomposition enables: 1) Explicit modeling of dominant evolving patterns through $\mathbf{\Phi}_{low}$, which exploits the long-term periodic and increment evolving patterns, i.e., targeting for *Challenge 1*; 2) Targeted regularization of uncerties via $\mathbf{R}_{high}$, i.e., targeting for *Challenge 2*. By jointly handling these regimes, the method enhances prediction stability against overfitting to domain-specific uncertainties while maintaining fidelity to long-term trends.

### 3.2.2 Robust Frequency Dynamics Learning

**Koopman Operator for Dominant Dynamics.** To model the dominant and temporally stable dynamics underlying the evolution of model parameters, we employ Koopman operator theory, which provides a linear surrogate for nonlinear dynamical systems by mapping them into a higher-dimensional functional space [20, 21]. Specifically, it assumes that the state can be mapped into a higher-dimensional Hilbert space via a measurement function $\varphi$, where the evolution is governed by an infinite-dimensional linear operator $\mathcal{K}$, such that: $\mathcal{K} \circ \varphi(x_t) = \varphi(\mathbf{\Phi}_{\boldsymbol{x}}(x_t)) = \varphi(x_{t+1})$, where $\mathbf{\Phi}$ is the unknown nonlinear transition function that governs the temporal evolution of the system [16]. Formally, let $\{\theta_{1,low}, \ldots, \theta_{T,low}\}$ denote the low-frequency components of model

parameters extracted via spectral decomposition. These components capture incremental and periodic variations that evolve gradually over time. We posit that this sequence admits a near-linear evolution in a latent space under a Koopman operator. Specifically, we seek an encoder $\phi : \mathbb{R}^D \to \mathbb{R}^m$ and a decoder $\phi^{-1} : \mathbb{R}^m \to \mathbb{R}^D$, such that the temporal evolution can be expressed as:

$$z_{t,low} = \phi(\theta_{t,low}), \ \hat{z}_{t+1,low} = \mathbf{K}z_{t,low}, \ \hat{\theta}_{t+1,low} = \phi^{-1}(\hat{z}_{t+1,low}), \tag{6}$$

where $\mathbf{K} \in \mathbb{R}^{m \times m}$ is a learnable linear transformation that approximates the finite-dimensional Koopman operator. To enforce temporal consistency in the predicted low-frequency dynamics, we define a reconstruction loss:

$$\mathcal{L}_{koop} = \sum_{t=1}^{T-1} \|z_{t+1,low} - \hat{z}_{t+1,low}\|_2^2, \tag{7}$$

which encourages accurate modeling of long-range parameter trends and mitigates error accumulation over time. This structure allows the model to forecast stable dynamics while preserving interpretability through linear temporal progression in the latent space.

**Regularization for High-Frequency Components.** While the low-frequency components capture dominant trends suitable for extrapolation via Koopman dynamics, the high-frequency components $\Theta_{high} = \{\theta_{1,high}, ..., \theta_{T,high}\}$ often represent transient noise or domain-specific uncertainties. Explicitly extrapolating these components could amplify noise and hinder generalization [15]. Therefore, instead of modeling forward dynamics for the high-frequency part, we focus on regularizing its behavior and incorporating its current state into the prediction. We use an encoder $\phi_{high} : \mathbb{R}^D \to \mathbb{R}^m$ to map the high-frequency component into the latent space: $z_{t,high} = \phi(\theta_{t,high})$, which captures the high-frequency information at time $t$. To mitigate overfitting to these potentially noisy high-frequency signals and encourage temporal smoothness in the underlying parameter trajectory, we impose the regularization on the sequence $\Theta_{high}$:

$$\mathcal{R}_{high} = \sum_{t=1}^{T-1} \|z_{t+1,high} - z_{t,high}\|_2^2. \tag{8}$$

This regularization encourages the high-frequency components to vary smoothly over time, implicitly suppressing sharp, transient changes that are less likely to generalize.

**Parameter Prediction and Joint Optimization.** To distinguish between high and low frequencies, we use separate encoders ($\phi_{low}, \phi_{high}$) for each frequency component, while sharing the same decoder ($\phi^{-1}$) to ensure consistency in the fusion of different components. To predict the parameters for the next timestamp, $\hat{\theta}_{t+1}$, we combine the extrapolated low-frequency latent with the regularized high-frequency latent and get the reconstructed result through the shared decoder:

---

**Algorithm 1** FreKoo End-to-End Learning Procedure

---

**Require:** Source domain datasets $D = \{D_1, ..., D_T\}$; Hyperparameters $\alpha, \beta, \gamma, \tau$; Koopman dimension $m$; Number of Epochs $N_{epochs}$.

**Ensure:** Trained FreKoo model components $(\Theta, \phi_{low}, \phi_{high}, \phi^{-1}, \mathbf{K})$ and target parameters $\hat{\theta}_{T+1}$.

1: Initialize base model $g(\cdot)$, Encoders $\phi_{low}, \phi_{high}$, Decoder $\phi^{-1}$, Koopman operator $\mathbf{K} \in \mathbb{R}^{m \times m}$.
2: **for** $epoch = 1$ **to** $N_{epochs}$ **do**
3:    **for** $t = 1$ **to** $T$ **do**
4:       Build base model $g(\cdot; \theta_t) : \mathcal{X}_t \to \mathcal{Y}_t \leftarrow \{\mathcal{D}_t\}$.
5:    **end for**
6:    Obtain parameters trajectory $\Theta = [\theta_1, ..., \theta_T]^T$.
7:    Compute DFT: $\Theta^f = \mathcal{F}(\Theta)$ via Eq. (2).
8:    Compute spectral magnitudes $\mathbf{M}_f$ via Eq. (3).
9:    Determine $Q_{top}$ based on $\tau$.
10:   Construct mask $\mathbf{B}$ via Eq. (4).
11:   Compute IDFT $\Theta_{low}$; $\Theta_{high}$ via Eq. (5).
12:   **for** $t = 1$ **to** $T - 1$ **do**
13:      $z_{t,low} \leftarrow \phi_{low}(\theta_{t,low})$,
14:      $\hat{\mathbf{z}}_{t+1,low} \leftarrow \mathbf{K}z_{t,low}$,
15:      $\hat{\theta}_{t+1,low} \leftarrow \phi^{-1}(\hat{z}_{t+1,low})$,
16:      $z_{t,high} \leftarrow \phi_{high}(\theta_{t,high})$,
17:      $\hat{\theta}_{t+1} \leftarrow \phi^{-1}(\mathbf{K}\phi_{low}(\theta_{t,low}) + \phi_{high}(\theta_{t,high}))$ via Eq. (9).
18:   **end for**
19: **end for**
20: Perform final spectral decomposition on learned $\Theta$ to get $\theta_{T,low}, \theta_{T,high}$.
21: $\hat{z}_{T+1,low} \leftarrow \mathbf{K}\phi_{low}(\theta_{T,low})$,
22: $z_{T,high} \leftarrow \phi_{high}(\theta_{T,high})$,
23: $\hat{\theta}_{T+1} \leftarrow \phi^{-1}(\hat{z}_{T+1,low} + z_{T,high})$.

---

$$\hat{\theta}_{t+1} = \phi^{-1}(\mathbf{K}\phi_{low}(\theta_{t,low}) + \phi_{high}(\theta_{t,high})). \tag{9}$$

This mechanism leverages the stability of the predicted low-frequency dynamics while incorporating the immediate context from the high-frequency component at time $t$. To ensure the overall predicted parameters align with the actual sequence, we further introduce a reconstruction error term:

$$\mathcal{L}_{rec} = \sum_{t=1}^{T-1} \|\theta_{t+1} - \hat{\theta}_{t+1}\|_2^2. \tag{10}$$

Therefore, the final objective function integrates the task loss $\mathcal{L}_{task} = \sum_{t=1}^{T} \ell(g(X_t; \theta_t), Y_t)$, the Koopman dynamics loss for low frequencies, the high-frequency smoothness regularization, and the overall parameter reconstruction loss:

$$\mathcal{L}_{total} = \mathcal{L}_{task} + \alpha\mathcal{L}_{rec} + \beta\mathcal{L}_{koop} + \gamma\mathcal{R}_{high}, \tag{11}$$

where $\alpha$, $\beta$, and $\gamma$ are hyperparameters controlling the relative contributions. This joint optimization framework encourages the model to learn a parameter trajectory $\Theta$ that not only performs well on historical tasks but also exhibits predictable low-frequency dynamics and smooth high-frequency variations, facilitating robust extrapolation to $\theta_{T+1}$ for the unseen future domain. The whole learning process is detailed in Algorithm 1.

## 3.3 Theoretical Insights

FreKoo's frequency-aware approach to TDG separates parameter dynamics into low-frequency ($\Theta_{low}$) for Koopman extrapolation and high-frequency ($\Theta_{high}$) for regularized smoothing. This strategy is theoretically grounded to enhance generalization to unseen future domains. We outline a generalization bound for the expected error on the target domain $D_{T+1}$. Let $\hat{\theta}_{T+1}$ be FreKoo's predicted parameter for $D_{T+1}$ (derived from Eq. (9)), and let $\theta_{T+1}^\star = \arg\min_\theta \mathbb{E}_{P_{T+1}}[\ell(g(X; \theta), Y)]$ be the optimal parameter under the target distribution $P_{T+1}$. Assume the loss $\ell$ is $L_\ell$-Lipschitz in its first argument, the base model $g(X; \theta)$ is $L_g$-Lipschitz w.r.t. $\theta$, and our encoders/decoder satisfy Lipschitz conditions (Assumption 1 in Appendix B.1). Further, assume the hypothesis class $\mathcal{G} = \{X \mapsto g(X; \phi^{-1}(z))\}$ has Rademacher complexity bounded by $C/\sqrt{n}$ (Assumption 2 in Appendix B.1), where $n$ is the target domain sample size.

**Theorem 1 (Multiscale Generalization Bound)** *Under the stated assumptions, with probability at least $1 - \delta > 0$, the expected excess risk on the target domain satisfies:*

$$\mathcal{E}_{T+1} := \mathbb{E}_{P_{T+1}}[\ell(g(X; \hat{\theta}_{T+1}), Y)] - \mathbb{E}_{P_{T+1}}[\ell(g(X; \theta_{T+1}^\star), Y)]$$
$$\leq L_\ell L_g L_{dec}(\mathcal{E}_{low} + \mathcal{E}_{high}) + \mathcal{O}(L_\ell L_g(\frac{C}{\sqrt{n}} + \sqrt{\frac{\log(1/\delta)}{n}})), \tag{12}$$

*where $\mathcal{E}_{low}$ and $\mathcal{E}_{high}$ are errors in the predicted low and high-frequency latent components.*

**Discussion.** Theorem 1 decomposes the excess risk into two primary factors:

*1) Parameter Prediction Error:* For the first term $L_\ell L_g L_{dec}(\mathcal{E}_{low} + \mathcal{E}_{high})$, the low-frequency error $\mathcal{E}_{low}$ is bounded by Koopman stability (Lemma 1), while the high-frequency error $\mathcal{E}_{high}$ is regulated via temporal smoothness (Lemma 2).

**Lemma 1 (Koopman Stability)** *For any initial low-frequency error $e_{t_0}$ and horizon $h = T + 1 - t_0$, $\|\mathbf{K}^h e_{t_0}\| \leq C_K(1+h)^{q-1}\|e_{t_0}\|$, where $q$ is the size of the largest Jordan block of $\mathbf{K}$ and $C_K = \kappa(V)$ for the Jordan basis $V$. If $\mathbf{K}$'s spectral radius $\rho(\mathbf{K}) < 1$, the bound sharpens to $C_K\rho(\mathbf{K})^h|e_{t_0}|$.*

**Lemma 2 (High-Frequency Smoothness Bias)** *Minimizing $\mathcal{R}_{high}$ is equivalent to maximum-a-posteriori estimation the Gaussian random-walk prior $z_{t+1,high} = z_{t,high} + \xi_t$, $\xi_t \sim \mathcal{N}(0, \sigma^2 I)$, with precision $\lambda = 1/(2\sigma^2)$.*

*2) Estimation Error:* The $\mathcal{O}(\cdot)$ term is a standard learning-theoretic bound related to hypothesis class complexity ($C$) and finite samples ($n$). Thus, FreKoo's distinct mechanisms, i.e., stable Koopman extrapolation for $\mathcal{E}_{low}$ and regularized smoothing for $\mathcal{E}_{high}$, directly address identifiable components of the generalization bound, promoting robust performance in evolving environments. A more precise statement and proof are provided in Appendix B.

# 4 Experiments

## 4.1 Experiment Settings

**Datasets.** Following [8], we evaluate FreKoo on seven datasets with various drift types. In classification, the synthetic Rotated-Moons and Rot-MNIST benchmarks create incremental drift through steadily increasing rotation angles, whereas the real-world streams ONP, Shuttle, and Elec2 exhibit incremental, periodic or unknown drifts, respectively. For regression, HousePrices and ApplianceEnergy also reflect a real-world non-stationary. These diverse datasets, featuring various drifts and real-world uncertainties, form a comprehensive test bed (details in Appendix C.1).

**Baselines.** We compare our method against four groups of baselines. 1) Time-Agnostic baselines: Offline, LastDomain and IncFinetune. 2) Continuous Domain Adaptation (CDA): CDOT [33], CIDA [22]. 3) TDG: GI [6], LSSAE [34], DDA [9], DRAIN [8]; 4) Continuous TDG: EvoS [35], Koodos [12]. Details can be found in Appendix C.2. All results were averaged over 5 independent runs, and we report the mean and standard deviation. The details of model implementations and hyper-parameters are detailed in Appendix C.3. The code is available at https://github.com/isenyu/FreKoo.

Table 1: Performance comparisons in terms of misclassification error (%) for classification and mean absolute error (MAE) for regression (both smaller the better).

| Methods | Classification | | | | | Regression | |
|---|---|---|---|---|---|---|---|
| | 2-Moons | Rot-MNIST | ONP | Shuttle | Elec2 | House | Appliance |
| Offline | $22.4 \pm 4.6$ | $18.6 \pm 4.0$ | $33.8 \pm 0.6$ | $0.77 \pm 0.10$ | $23.0 \pm 3.1$ | $11.0 \pm 0.36$ | $10.2 \pm 1.1$ |
| LastDomain | $14.9 \pm 0.9$ | $17.2 \pm 3.1$ | $36.0 \pm 0.2$ | $0.91 \pm 0.18$ | $25.8 \pm 0.6$ | $10.3 \pm 0.16$ | $9.1 \pm 0.7$ |
| IncFinetune | $16.7 \pm 3.4$ | $10.1 \pm 0.8$ | $34.0 \pm 0.3$ | $0.83 \pm 0.07$ | $27.3 \pm 4.2$ | $9.7 \pm 0.01$ | $8.9 \pm 0.5$ |
| CDOT [33] | $9.3 \pm 1.0$ | $14.2 \pm 1.0$ | $34.1 \pm 0.0$ | $0.94 \pm 0.17$ | $17.8 \pm 0.6$ | - | - |
| CIDA [22] | $10.8 \pm 1.6$ | $9.3 \pm 0.7$ | $34.7 \pm 0.6$ | - | $14.1 \pm 0.2$ | $9.7 \pm 0.06$ | $8.7 \pm 0.2$ |
| GI [6] | $3.5 \pm 1.4$ | $7.7 \pm 1.3$ | $36.4 \pm 0.8$ | $0.29 \pm 0.05$ | $16.9 \pm 0.7$ | $9.6 \pm 0.02$ | $8.2 \pm 0.6$ |
| LSSAE [34] | $9.9 \pm 1.1$ | $9.8 \pm 3.6$ | $38.8 \pm 1.1$ | $0.22 \pm 0.01$ | $16.1 \pm 1.4$ | - | - |
| DDA [9] | $9.7 \pm 1.5$ | $7.6 \pm 0.7$ | $34.0 \pm 0.3$ | $\underline{0.21 \pm 0.02}$ | $12.8 \pm 1.1$ | $9.5 \pm 0.12$ | $6.1 \pm 0.1$ |
| DRAIN [8] | $3.2 \pm 1.2$ | $7.5 \pm 1.1$ | $38.3 \pm 1.2$ | $0.26 \pm 0.05$ | $12.7 \pm 0.8$ | $9.3 \pm 0.14$ | $6.4 \pm 0.4$ |
| EvoS [35] | $3.0 \pm 0.4$ | $7.3 \pm 0.6$ | $35.4 \pm 0.2$ | $0.23 \pm 0.01$ | $11.8 \pm 0.5$ | $9.8 \pm 0.10$ | $7.2 \pm 0.1$ |
| Koodos [12] | $\underline{1.3 \pm 0.4}$ | $7.0 \pm 0.3$ | $\underline{33.5 \pm 0.4}$ | $0.24 \pm 0.04$ | - | $\mathbf{8.8 \pm 0.19}$ | $\underline{4.8 \pm 0.3}$ |
| FreKoo (ours) | $\mathbf{1.0 \pm 0.3}$ | $\mathbf{6.9 \pm 0.7}$ | $\mathbf{32.3 \pm 0.3}$ | $\mathbf{0.20 \pm 0.02}$ | $\mathbf{9.2 \pm 0.7}$ | $\underline{9.0 \pm 0.11}$ | $\mathbf{4.0 \pm 0.1}$ |

## 4.2 Results and Analysis

As shown in Table 1, our proposed FreKoo achieves state-of-the-art performance on six out of seven TDG benchmarks, significantly surpassing prior methods across both classification and regression tasks. FreKoo shows notable improvements on datasets characterized by synthetic incremental drifts, such as 2-Moons and Rot-MNIST, where it attains the lowest error rates. Crucially, it excels particularly on benchmarks involving periodic drifts, like Elec2 and Appliance, achieving marked effectiveness with error rates of 9.2% and 4.0 MAE, respectively. This highlights the strength of our frequency-domain approach in capturing and leveraging underlying periodic dynamics, solving *Challenge 1*. Furthermore, FreKoo's consistent leading performance across diverse real-world datasets (e.g., ONP, Elec2, Shuttle, Appliance) underscores its robustness against overfitting to domain-specific artifacts or local noise. This resilience stems from its dual mechanism: employing the Koopman operator to model stable low-frequency dynamics (trends/periodicity) while simultaneously regularizing high-frequency components to mitigate noise and prevent adaptation to transient, domain-specific characteristics, solving *Challenge 2*. Therefore, these results validate the efficacy of FreKoo's frequency-Koopman design for robustly handling complex concept drifts, while maintaining generalization capability in real-world dynamic environments. We also provide the visualization of the decision boundary and analysis compared with SOTA methods in Appendix D.2.

## 4.3 Ablation Study

As shown in Table 2, our ablation study on diverse drift types (e.g., incremental drift on Rotated Moons, periodic drift on Elec2 and Appliance) systematically evaluates FreKoo's core components. Removing the Koopman operator (**w/o Koop.**) severely degrades performance, underscoring its

criticality for modeling parameter dynamics. Similarly, omitting frequency decomposition (**w/o Freq.**) significantly harms performance on periodic datasets (Elec2, Appliance), confirming spectral analysis is vital for disentangling stable low-frequency trends from high-frequency noise, especially for complex periodic patterns.

In addition, we further analyze the training objectives (Eq. (11)) by removing individual loss terms to highlight the importance of each component. Removing the overall parameter reconstruction loss (**w/o** $\mathcal{L}_{rec}$, Eq. (10)) decouples the learned latent dynamics from the final parameter prediction task and thus hurts fidelity. Excluding the Koopman latent consistency loss (**w/o** $\mathcal{L}_{koop}$, Eq. (7)) weakens the learned linear dynamics (**K**), impairing the reliable extrapolation of stable trends. Disabling the high-frequency smoothing regularization (**w/o** $\mathcal{R}_{high}$, Eq. (8)) increases sensitivity

Table 2: Ablation study. Comparison between FreKoo and its variants across two datasets for classification and one dataset for regression.

| Variants | 2-Moons | Elec2 | Appliance |
|---|---|---|---|
| w/o Koop. | $15.3 \pm 1.7$ | $28.9 \pm 2.4$ | $9.4 \pm 0.7$ |
| w/o Freq. | $2.6 \pm 0.8$ | $14.2 \pm 1.9$ | $5.2 \pm 0.4$ |
| w/o $\mathcal{L}_{rec}$ | $9.1 \pm 1.7$ | $12.2 \pm 1.0$ | $4.3 \pm 0.4$ |
| w/o $\mathcal{L}_{koop}$ | $6.7 \pm 0.9$ | $10.0 \pm 0.8$ | $4.2 \pm 0.6$ |
| w/o $\mathcal{R}_{high}$ | $2.3 \pm 0.6$ | $10.5 \pm 1.7$ | $4.2 \pm 0.3$ |
| FreKoo | $\mathbf{1.0 \pm 0.3}$ | $\mathbf{9.2 \pm 0.7}$ | $\mathbf{4.0 \pm 0.1}$ |

to domain-specific noise and transient artifacts by allowing unfiltered high-frequency fluctuations ($z_{t,high}$) into the prediction (Eq. (9)). These results collectively affirm that FreKoo's spectral decomposition, Koopman modeling, high-frequency regularization, and associated losses are all essential for robust temporal domain generalization.

## 4.4 Periodicity Modeling Performance

To rigorously evaluate long-term periodicity handling, we extended the 10-domain rotating 2-Moons benchmark (0-9 domains, 18° increments) into a 37-domain sequence. This sequence embeds recurring cycles following the conceptual pattern $\{0 \rightarrow 9 \rightarrow 0 \rightarrow 9 \rightarrow 0\}$. Training on the first 36 domains and testing on the 37th forces models to exploit long-range pattern recurrence. We compare FreKoo against GI [6], DRAIN [8], and Koodos [12]. As shown in Figure 3a, FreKoo significantly outperforms these methods on this periodic benchmark. This superior result demonstrates FreKoo's specific ability to capture and utilize the explicitly embedded long-term evolutions, validating its robustness and enhanced generalization capabilities in real-world data streams with complex drifts.

In addition, Figure 3b qualitatively illustrates FreKoo's parameter evolution: the low-frequency component ($\Theta_{low}$) effectively captures the underlying periodic trend of the raw parameters ($\Theta$), while the high-frequency component ($\Theta_{high}$) isolates transient variations. The reconstructed parameter

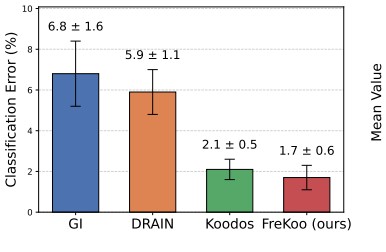
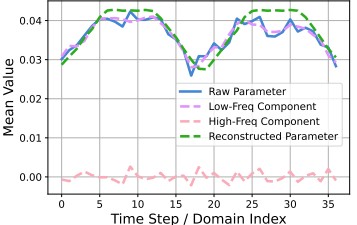

(a) Quantitative performance      (b) Qualitative performance

Figure 3: Periodicity modeling performance on P-Moons dataset.

($\hat{\Theta}$) tracks these long-term evolutions smoothly. Such targeted spectral separation and dual modeling enable FreKoo to learn more stable parameter evolutions, mitigating overfitting to transient details and fostering improved generalization. A more extensive qualitative analysis is shown in Appendix D.3.

## 4.5 Sensitivity Analysis

We analyzed sensitivity to the spectral energy preservation ratio $\tau$ and loss weights $\alpha, \beta, \gamma$ on 2-Moons with synthesized incremental drift and Appliance with real-world periodicity and uncertainties. As shown in Figure 4a, higher $\tau$ benefits 2-Moons, while an intermediate $\tau$ is optimal for Appliance. This underscores $\tau$'s crucial role in balancing the preservation of essential low-frequency dynamics against filtering disruptive high-frequency noise, particularly for complex noisy real-world data. For $\alpha, \beta, \gamma$, grid search over [0.01, 0.1, 1, 10, 100] (Figure 4b- 4d) confirmed that optimal performance hinges on balancing reconstruction accuracy, Koopman stability, and high-frequency smoothing.

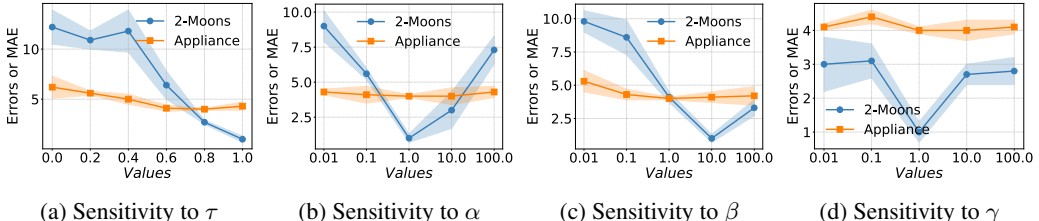

Figure 4: Parameters sensitivity on 2-Moons and Appliance datasets.

# 5    Conclusion and Limitations

This paper tackled key challenges of TDG, i.e., modeling long-term periodicity and achieving robustness against domain-specific uncertainties. Our proposed FreKoo introduces a novel frequency-aware perspective. It analyzes parameter trajectories in the frequency domain, disentangling low-frequency dynamics modeled via the Koopman operator from high-frequency noise smoothed via targeted regularization. Both theoretical analysis and extensive experiments validate FreKoo's effectiveness, demonstrating its significant generalization capabilities and robustness under complex concept drifts.

A limitation of our work is the reliance on a pre-defined heuristic for frequency separation, rather than an adaptive mechanism that learns the optimal threshold from data. Furthermore, the current FreKoo framework is not designed for fully online or continuous settings. This restricts its applicability in real-world deployment scenarios that require continuous model adaptation to dynamic data streams.

## Acknowledgments and Disclosure of Funding

The work was supported by the Australian Research Council (ARC) under Laureate project FL190100149 and Discovery Project DP220102635.

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

# Appendix

## A    Additional Related Works

**Domain Adaptation/Generalization.** Domain Adaptation (DA) requires access to both source and target domain data during training, employing methods like domain-invariant learning, domain mapping , and ensemble approaches to minimize domain discrepancy [36–38]. Domain Generalization (DG) aims to train a model on multiple source domains to generalize to unseen target domains without access to target data, using techniques such as data augmentation, representation learning, and meta-learning [39, 40]. Both DG and DA frameworks, by treating domains as independent entities, often fail to account for the smooth evolutionary patterns present in time-ordered domain sequences.

**Real-world Concept Drift.** Concept drift signifies a change in the underlying data distribution over time, formally occurring between time $t$ and $t + 1$ if the joint distribution $P_{t+1}(X, y) \neq P_t(X, y)$. This non-stationarity poses a fundamental challenge, requiring models to adapt dynamically to maintain predictive performance and reliability [3, 13, 23]. While concept drift manifests in various forms, including sudden, gradual, incremental, and recurring (periodic) patterns, much prior research treats it as unpredictable, often employing detect-then-adapt strategies [24, 25, 41]. In contrast, the Temporal Domain Generalization (TDG) setting frequently focuses on leveraging more predictable evolutionary dynamics for proactive generalization [8]. However, existing TDG methods often oversimplify these dynamics, primarily modeling smooth, incremental drifts while struggling to capture complex, long-range temporal structures like periodicity (a key limitation, Challenge 1) which are prevalent in many real-world data streams yet demand different modeling approaches than simple monotonic progression.

Furthermore, a common methodology within TDG involves segmenting the continuous data stream into discrete temporal domains $\{D_1, D_2, \ldots, D_T\}$, assuming concept drift occurs primarily between these sequential domains [6]. This discretization, while simplifying the problem, often struggles to perfectly align with real-world data streams, potentially violating the implicit assumption that data within each domain $D_t$ is Independent and Identically Distributed (IID). Consequently, individual domains may harbor internal non-IID structures or localized noise patterns [42]. The presence of intra-domain complexities leads to sensitivity to domain-specific noise and artifacts, hindering robust generalization (a second key limitation, Challenge 2), particularly for TDG models designed predominantly to address shifts occurring only at the domain boundaries. Effectively handling both the diverse inter-domain evolutionary dynamics (including periodicity) and these intra-domain data characteristics is therefore crucial for robust temporal generalization.

**Dynamics Learning via Koopman Theory.** Koopman operator theory [20] offers a powerful framework for analyzing nonlinear dynamical systems via linear operators acting on observables in an infinite-dimensional function space. In practice, machine learning methods approximate this by learning transformations (often using autoencoders) into a latent Koopman space where the system's dynamics can be linearly propagated [43, 16]. This learned linear structure facilitates efficient long-term prediction and control [44]. While Koopman-based approaches have seen increasing use in time series forecasting [31], they typically focus on modeling the dynamics within a given data stream (data-centric). Our work diverges by applying Koopman theory to model the evolution of model parameters, interpreting parameter changes under concept drift. Furthermore, we uniquely integrate this parameter-centric Koopman modeling with frequency-domain analysis for disentangling dynamics, which towards stable and robust generalization in complex drifting scenarios.

## B    Theoretical Analysis Details

This section provides supplementary details for the theoretical analysis presented in Section 3.3, including formal assumptions, detailed statements of the theorem and lemmas, and proofs.

### B.1    Assumptions

**Assumption 1 (Encoder/Decoder Lipschitz)** *The encoders* $\phi_{low} : \mathbb{R}^D \to \mathbb{R}^m$ *and* $\phi_{high} : \mathbb{R}^D \to \mathbb{R}^m$ *are* $L_\phi$-*Lipschitz:*

$$\forall \theta, \theta' \in \mathbb{R}^D, \quad \|\phi(\theta) - \phi(\theta')\| \leq L_\phi \|\theta - \theta'\|, \quad for \ \phi \in \{\phi_{low}, \phi_{high}\}. \tag{13}$$

The shared decoder $\phi^{-1} : \mathbb{R}^m \to \mathbb{R}^D$ is $L_{dec}$-Lipschitz:

$$\forall z, z' \in \mathbb{R}^m, \quad \left\| \phi^{-1}(z) - \phi^{-1}(z') \right\| \leq L_{dec} \left\| z - z' \right\|. \tag{14}$$

This implies that small changes in the parameter space lead to bounded changes in the latent space, and vice-versa, ensuring stability of the transformations.

**Assumption 2 (Model-class Capacity)** *The composite hypothesis class $\mathcal{G} = \{x \mapsto g(x; \phi^{-1}(z)) \mid z \in \mathbb{R}^m\}$, where $g$ is the base model and $\phi^{-1}$ is the decoder, has an empirical Rademacher complexity $\mathcal{R}_n(\mathcal{G})$ on any sequence of $n$ samples drawn from a target domain $D_{T+1}$ (which is $\beta$-mixing [45, 46]) bounded as:*

$$\mathcal{R}_n(\mathcal{G}) \leq \frac{C}{\sqrt{n}}, \tag{15}$$

For constant $C > 0$, this assumption bounds the complexity of the function class our model can represent, crucial for generalization guarantees.

## B.2 Detailed Lemma 1 and Proof

**Lemma 3 (Restatement of Lemma 1 - Koopman Stability)** *For any initial low-frequency latent error $e_{t_0,low} = z_{t_0,low} - \hat{z}_{t_0,low}$ at an initial time $t_0$, and a prediction horizon $h = (T+1) - t_0$, the propagated error $e_{t_0+h,low} = z_{t_0+h,low} - \hat{z}_{t_0+h,low}$ under the learned Koopman operator $K \in \mathbb{R}^{m \times m}$ is bounded. Specifically,*

$$\left\| K^h e_{t_0,low} \right\| \leq C_K (1+h)^{q-1} \left\| e_{t_0,low} \right\|, \tag{16}$$

*where $q$ is the size of the largest Jordan block of $K$, and $C_K = \kappa(V) = \|V\| \|V^{-1}\|$ is the condition number of the matrix $V$ from the Jordan decomposition $K = VJV^{-1}$. If $K$ is diagonalizable (i.e., $q = 1$) and its spectral radius $\rho(K) = \max_i |\lambda_i(K)| < 1$, the bound sharpens to:*

$$\left\| K^h e_{t_0,low} \right\| \leq \kappa(V) \rho(K)^h \left\| e_{t_0,low} \right\|. \tag{17}$$

**Proof 1 (Proof of Lemma 3)** *Jordan Canonical Form: Any square matrix $K \in \mathbb{R}^{m \times m}$ admits a Jordan decomposition $K = VJV^{-1}$, where $J = diag(J_1, J_2, \ldots, J_s)$ is a block diagonal matrix. Each $J_i$ is a Jordan block corresponding to an eigenvalue $\lambda_i$ of $K$. $V$ is the matrix whose columns are the eigenvectors and generalized eigenvectors of $K$. Let $q$ be the size of the largest Jordan block.*

**Power of a Matrix via Jordan Form:** *The $h$-th power of $K$ is $K^h = (VJV^{-1})^h = VJ^hV^{-1}$. Consequently, $J^h = diag(J_1^h, J_2^h, \ldots, J_s^h)$.*

**Bound on the Norm of $J_i^h$:** *For a Jordan block $J_i$ of size $q_i$ with eigenvalue $\lambda_i$, and assuming $|\lambda_i| \leq 1$ for stability, a standard result [47] states that there exists a constant $C'_{q_i}$ such that for $h \geq 0$, we have,*

$$\left\| J_i^h \right\| \leq C'_{q_i} (1+h)^{q_i-1} |\lambda_i|^{h-(q_i-1)} \leq C'_{q_i} (1+h)^{q_i-1} \quad (\text{if } |\lambda_i| \leq 1). \tag{18}$$

*Thus, for the overall Jordan form matrix $J$, if $q = \max_i q_i$:*

$$\left\| J^h \right\| \leq C''(1+h)^{q-1}, \tag{19}$$

*for some constant $C''$ that depends on the constants $C'_{q_i}$ and the number of blocks.*

**Error Propagation Bound:** *The propagated error is $K^h e_{t_0,low}$. Its norm is:*

$$\left\| K^h e_{t_0,low} \right\| = \left\| VJ^h V^{-1} e_{t_0,low} \right\| \tag{20}$$

$$\leq \|V\| \left\| J^h \right\| \left\| V^{-1} \right\| \left\| e_{t_0,low} \right\| \tag{21}$$

$$\leq \kappa(V) C''(1+h)^{q-1} \left\| e_{t_0,low} \right\|, \tag{22}$$

*where $\kappa(V) = \|V\| \|V^{-1}\|$ is the condition number of $V$. We define $C_K = \kappa(V)C''$ (or absorb $C''$ into the definition presented in the lemma statement). This shows polynomial growth if $q > 1$.*

**Diagonalizable Case:** *If $K$ is diagonalizable, then $J$ is a diagonal matrix of eigenvalues, $J = \Lambda = diag(\lambda_1, \ldots, \lambda_m)$. Then $J^h = diag(\lambda_1^h, \ldots, \lambda_m^h)$.*

$$\left\| J^h \right\| = \max_i \left| \lambda_i^h \right| = (\max_i |\lambda_i|)^h = \rho(K)^h. \tag{23}$$

*In this case, the error bound becomes:*

$$\left\| K^h e_{t_0,low} \right\| \leq \kappa(V)\rho(K)^h \left\| e_{t_0,low} \right\|. \tag{24}$$

*If $\rho(K) < 1$, this implies exponential decay of the error, which is highly desirable for the stability of low-frequency dynamics. The loss term $\mathcal{L}_{koop}$ encourages learning such a stable $K$.*

**Discussion.** We focus on the propagation of the low-frequency latent error $e_{t,low} = \hat{z}_{t,low} - z^*_{t,low}$. By recursively analyzing the single-step error recurrence $e_{t+1,low} = Ke_{t,low} + e_{t,low}$, where $e_{t,low} = Kz^*_{t,low} - z^*_{t+1,low}$ is the single-step error, we can express it after an $l$-step prediction as: $e_{t_0+l,low} = K^l e_{t_0,low} + \sum_{i=0}^{l-1} K^i e_{t_0+l-1-i,low}$. Taking the norm, we get the multi-step error bound: $\|e_{t_0+l,low}\| \leq \|K^l\| \cdot \|e_{t_0,low}\| + \sum_{i=0}^{l-1} \|K^i\| \cdot \|e_{t_0+l-1-i,low}\|$. This reveals that the final error is not only dependent on the initial error ($e_{t_0,low}$), but also on the sum of all previous errors ($e_{t,low}$) accumulated along the entire prediction path. This provides a stronger justification for our full-trajectory optimization approach. The stability of $K$ bounds $\|K^i\|$ and controls the propagation of intermediate errors. The objective function minimizes losses across all domains, directly works to reduce the magnitude of each single-step error $\|e_{t,low}\|$.

## B.3 Detailed Lemma 2 and Proof

**Lemma 4 (Restatement of Lemma 2 - High-Frequency Smoothness Bias)** *Minimizing the high-frequency regularization term $\mathcal{R}_{high} = \sum_{t=1}^{T-1} \|z_{t+1,high} - z_{t,high}\|_2^2$ is equivalent to performing Maximum A Posteriori (MAP) estimation for the latent high-frequency sequence $Z_{high} = \{z_{1,high}, \ldots, z_{T,high}\}$ under a Gaussian random-walk prior $p(z_{t+1,high}|z_{t,high}) = \mathcal{N}(z_{t+1,high}|z_{t,high}, \sigma^2 I)$. The precision of this prior is $\lambda_{prior} = 1/(2\sigma^2)$.*

**Proof 2 (Proof of Lemma 4)** *Gaussian Random-Walk Prior: Assume the high-frequency latent states $z_{t,high}$ evolve according to a first-order Markov process:*

$$z_{t+1,high} = z_{t,high} + \xi_t, \quad where \ \xi_t \sim \mathcal{N}(0, \sigma^2 I), \tag{25}$$

*and $\xi_t$ are Gaussian noise vectors. The conditional probability (prior) is:*

$$p(z_{t+1,high}|z_{t,high}) = \frac{1}{(2\pi\sigma^2)^{m/2}} \exp\left(-\frac{1}{2\sigma^2} \|z_{t+1,high} - z_{t,high}\|_2^2\right), \tag{26}$$

*Log-Likelihood of the Sequence under Prior: For a sequence $Z_{high} = \{z_{1,high}, \ldots, z_{T,high}\}$, the joint probability under this prior model is $p(Z_{high}) = p(z_{1,high}) \prod_{t=1}^{T-1} p(z_{t+1,high}|z_{t,high})$. The log-probability is:*

$$\log p(Z_{high}) = \log p(z_{1,high}) + \sum_{t=1}^{T-1} \log p(z_{t+1,high}|z_{t,high}) \tag{27}$$

$$= \log p(z_{1,high}) + \sum_{t=1}^{T-1} \left[-\frac{m}{2} \log(2\pi\sigma^2) - \frac{1}{2\sigma^2} \|z_{t+1,high} - z_{t,high}\|_2^2\right]. \tag{28}$$

*Maximum A Posteriori (MAP) Estimation: MAP estimation seeks to find $Z_{high}$ that maximizes $p(Z_{high})$. This is equivalent to maximizing $\log p(Z_{high})$, or equivalently, minimizing $-\log p(Z_{high})$:*

$$\hat{Z}_{high}^{MAP} = \arg\min_{Z_{high}} \left[-\log p(z_{1,high}) + \sum_{t=1}^{T-1} \left(\frac{m}{2} \log(2\pi\sigma^2) + \frac{1}{2\sigma^2} \|z_{t+1,high} - z_{t,high}\|_2^2\right)\right]. \tag{29}$$

*The terms $-\log p(z_{1,high})$ and $\frac{m}{2} \log(2\pi\sigma^2)$ are constant with respect to the sum of squared differences. Thus, the optimization simplifies to:*

$$\hat{Z}_{high}^{MAP} = \arg\min_{Z_{high}} \sum_{t=1}^{T-1} \frac{1}{2\sigma^2} \|z_{t+1,high} - z_{t,high}\|_2^2. \tag{30}$$

***Equivalence to*** $\mathcal{R}_{high}$***:*** *The regularization term is* $\mathcal{R}_{high} = \sum_{t=1}^{T-1} \|z_{t+1,high} - z_{t,high}\|_2^2$*. Minimizing* $\mathcal{R}_{high}$ *is equivalent to minimizing* $\frac{1}{2\sigma^2}\mathcal{R}_{high}$*, since* $\frac{1}{2\sigma^2}$ *is a positive constant. This shows that minimizing* $\mathcal{R}_{high}$ *is equivalent to MAP estimation under the specified Gaussian random-walk prior with precision* $\lambda_{prior} = 1/(2\sigma^2)$*. This encourages smoothness in the trajectory of high-frequency components.*

## B.4  Detailed Proof of Theorem 1

**Theorem 2 (Restatement of Theorem 1 - Multiscale Generalization Bound)** *Let* $\hat{\theta}_{T+1}$ *be the predicted parameter for the target domain* $D_{T+1}$ *obtained via Eq. (9), and let* $\theta_{T+1}^* = \arg\min_\theta \mathbb{E}_{P_{T+1}}[\ell(g(X;\theta),Y)]$ *be the optimal parameter for the target distribution* $P_{T+1}$*. The loss function* $\ell(u,y)$ *is* $L_\ell$*-Lipschitz in its first argument* $u$*. The base model* $g(x;\theta)$ *is* $L_g$*-Lipschitz with respect to its parameters* $\theta$*. Under Assumptions 1 (Lipschitz encoders/decoder) and 2 (bounded Rademacher complexity C), the expected target-domain excess risk* $E_{risk} := \mathbb{E}_{P_{T+1}}[\ell(g(X;\hat{\theta}_{T+1}),Y)] - \mathbb{E}_{P_{T+1}}[\ell(g(X;\theta_{T+1}^*),Y)]$ *satisfies with probability at least* $1 - \delta > 0$*:*

$$E_{risk} \le L_\ell L_g L_{dec}(\mathcal{E}_{low} + \mathcal{E}_{high}) + 2L_\ell L_g \frac{C}{\sqrt{n}} + L_\ell L_g \sqrt{\frac{B^2 \log(1/\delta)}{2n}}, \tag{31}$$

*where* $\mathcal{E}_{low} = \left\|\hat{z}_{T+1,low} - z_{T+1,low}^*\right\|$ *is the error in the predicted low-frequency latent component (bounded by Lemma 3),* $\mathcal{E}_{high} = \left\|\hat{z}_{T+1,high} - z_{T+1,high}^*\right\|$ *is the error in the high-frequency latent component (regularized by* $\mathcal{R}_{high}$ *as per Lemma 4),* $n_T$ *is the number of samples in the target domain, and B is an upper bound on* $L_g L_{dec}\|z - z'\|$*.*

**Proof 3 (Proof of Theorem 2)** *The proof involves combining bounds on parameter prediction error with standard generalization theory.*

***Bounding Excess Risk by Parameter Error:*** *The excess risk can be related to the difference in parameters using Lipschitz properties:*

$$\begin{aligned}
E_{risk} &= \mathbb{E}_{P_{T+1}}[\ell(g(X;\hat{\theta}_{T+1}),Y) - \ell(g(X;\theta_{T+1}^*),Y)] \\
&\le \mathbb{E}_{P_{T+1}}[L_\ell \left|g(X;\hat{\theta}_{T+1}) - g(X;\theta_{T+1}^*)\right|] \quad \textit{(by } L_\ell\textit{-Lipschitz of } \ell\textit{)} \\
&\le L_\ell \mathbb{E}_{P_{T+1}}[L_g \left\|\hat{\theta}_{T+1} - \theta_{T+1}^*\right\|] \quad \textit{(by } L_g\textit{-Lipschitz of } g\textit{)} \\
&= L_\ell L_g \left\|\hat{\theta}_{T+1} - \theta_{T+1}^*\right\|.
\end{aligned} \tag{32}$$

*Since* $\hat{\theta}_{T+1} = \phi^{-1}(\hat{z}_{T+1})$ *and* $\theta_{T+1}^* = \phi^{-1}(z_{T+1}^*)$ *by* $L_{dec}$*-Lipschitz of* $\phi^{-1}$ *(Assumption 1), we have:*

$$\left\|\hat{\theta}_{T+1} - \theta_{T+1}^*\right\| \le L_{dec} \left\|\hat{z}_{T+1} - z_{T+1}^*\right\|. \tag{33}$$

*The latent state* $z = z_{low} + z_{high}$*. So,* $\hat{z}_{T+1} = \hat{z}_{T+1,low} + \hat{z}_{T+1,high}$*.*

$$\begin{aligned}
\left\|\hat{z}_{T+1} - z_{T+1}^*\right\| &= \left\|(\hat{z}_{T+1,low} - z_{T+1,low}^*) + (\hat{z}_{T+1,high} - z_{T+1,high}^*)\right\| \\
&\le \left\|\hat{z}_{T+1,low} - z_{T+1,low}^*\right\| + \left\|\hat{z}_{T+1,high} - z_{T+1,high}^*\right\| \\
&= \mathcal{E}_{low} + \mathcal{E}_{high}.
\end{aligned} \tag{34}$$

*Therefore, the first term of the risk bound related to parameter prediction error is:*

$$PredictionErrorTerm \le L_\ell L_g L_{dec}(\mathcal{E}_{low} + \mathcal{E}_{high}). \tag{35}$$

$\mathcal{E}_{low}$ *is bounded by Lemma 3 and* $\mathcal{E}_{high}$ *is controlled by the regularization from Lemma 4.*

***Standard Generalization Error Bound (Estimation Error):*** *The previous term relates the risk of our predictor* $\hat{\theta}_{T+1}$ *to the risk of the optimal* $\theta_{T+1}^*$*, assuming* $\theta_{T+1}^*$ *is within our hypothesis class. We also need to account for the generalization error from the empirical risk minimizer (over the unseen target domain* $D_{T+1}$*) to the true risk. Let* $\mathcal{L}(\theta) = \ell(g(X;\theta),Y)$*. The composite function* $X \mapsto \mathcal{L}(\theta)$

is $L_{\mathcal{L}}$-Lipschitz where $L_{\mathcal{L}} \approx L_l L_g$. Using a standard Rademacher complexity-based generalization bound [45]: With probability at least $1 - \delta$ over the draw of $n$ samples for $D_{T+1}$:

$$\sup_{\theta \in \mathcal{H}_\Theta} (\mathbb{E}_{P_{T+1}}[\mathcal{L}(\theta)] - \mathbb{E}_n[\mathcal{L}(\theta)]) \leq 2L_l L_g \mathcal{R}_n(\mathcal{G}_\Theta) + L_l L_g \sqrt{\frac{B_\Theta^2 \log(1/\delta)}{2n}}, \qquad (36)$$

where $\mathcal{H}_\Theta$ is the space of parameters, $\mathcal{G}_\Theta$ is the function class induced by $g(X; \theta)$, $\mathcal{R}_n(\mathcal{G}_\Theta)$ is its Rademacher complexity (here using $\mathcal{R}_{n_T}(\mathcal{G}) \leq C/\sqrt{n}$ from Assumption 2 where $\mathcal{G}$ is the class $g(X; \phi^{-1}(z))$), and $B_\Theta$ is a bound on $g(X; \theta)$. More directly, the generalization gap for our specific predictor $\hat{\theta}_{T+1}$ (derived from $z \in \mathbb{R}^m$) is:

$$\mathbb{E}_{P_{T+1}}[\ell(g(X; \hat{\theta}_{T+1}), Y)] \leq \mathbb{E}_n[\ell(g(X; \hat{\theta}_{T+1}), Y)] + 2L_\ell L_g \frac{C}{\sqrt{n}} + L_\ell L_g \sqrt{\frac{B^2 \log(1/\delta)}{2n}}, \quad (37)$$

where $B$ bounds $L_{dec} \|z\|$. This term essentially bounds how much the true risk of $\hat{\theta}_{T+1}$ can deviate from its (unobserved) empirical risk on $D_{T+1}$. The excess risk definition compares true risks. The parameter error term in Eq. (35) captures the suboptimality of $\hat{\theta}_{T+1}$ relative to $\theta^*_{T+1}$. The estimation error term often appears when bounding $E_P[l(\hat{\theta}_{ERM})] - E_P[l(\theta^*)]$. The provided bound in Eq. (31) represents a structure where the first term is the approximation error (how far our best hypothesis $\hat{\theta}_{T+1}$ is from $\theta^*_{T+1}$ in terms of risk, scaled by Lipschitz constants) and the second and third terms represent the estimation error (how well one can estimate the risk of any hypothesis in the class $\mathcal{G}$ from $n$ samples).

**Combining Terms:** The bound structure in our method: $E_{risk} \leq PredictionErrorTerm + EstimationError$. The Prediction Error is $L_l L_g L_{dec}(\mathcal{E}_{low} + \mathcal{E}_{high})$. The Estimation Error, using Assumption 2 and standard results for Lipschitz losses, can be written as $2L_\ell L_g \mathcal{R}_n(\mathcal{G}) + ConfidenceTerm$. Substituting $\mathcal{R}_n(\mathcal{G}) \leq C/\sqrt{n}$: $EstimationError \approx 2L_\ell L_g \frac{C}{\sqrt{n}} + L_\ell L_g \sqrt{\frac{B^2 \log(1/\delta)}{2n}}$. Combining these yields Eq. (31).

## C   Experiment Settings

### C.1   Datasets

**Rotated 2 Moons.** This benchmark adapts the 2-Moons dataset to model concept drift via rotation. It contains 1,800 2-dimensional samples across two classes, divided into 10 sequential domains. Each domain is rotated $18°$ counter-clockwise relative to the previous one. We train on domains 0-8 and test on domain 9, where the drift is caused by the incremental rotation.

**Rotated MNIST.** We randomly sampled 1000 instances from the MNIST dataset and constructed a total of five domains by successively rotating them counter-clockwise by $15°$, analogous to the Rotated 2 Moons setup. The first four rotated domains are used for training, while the fifth domain serves as the test set, creating incremental drift induced by progressive rotation transformations.

**Online News Popularity (ONP)[2].** This dataset aggregates heterogeneous features of articles published by Mashable over two years, aiming to predict social media shares (popularity). It comprises 39,797 samples with 58 features, where concept drift is characterized by temporal shifts in popularity patterns. We partition the data into 6 time-ordered domains, using the first five for training and the last for testing. The dataset undergoes slight real-world concept drift over the observed time period.

**Shuttle[3].** The Shuttle dataset contains 58,000 instances of multi-class flight status classification under severe class imbalance. It is partitioned into 8 time-stamped domains using a chronological split: domains spanning timestamps 30-70 serve as training data, while the most recent period (70–80) is reserved for testing. The dataset also has real-world concept drifts over the observed time period.

**Electrical Demand[4].** This dataset records electricity demand in a province, addressing a binary classification task to predict whether 30-minute demand exceeds or falls below the daily average for

---

[2]https://archive.ics.uci.edu/dataset/332/online+news+popularity

[3]https://archive.ics.uci.edu/dataset/148/statlog+shuttle

[4]https://web.archive.org/web/20191121102533/http://www.inescporto.pt/j̃gama/ales/ales_5.html

that time period. After removing instances with missing values, it contains 28,222 samples with 8 features. It is partitioned into 30 two-week chronological domains, with the first 29 used for training and the 30th for testing. Seasonal variations in demand induce concept drift, making this a real-world benchmark capturing both periodic and incremental drift patterns.

**House Prices**[5]**.** This dataset comprises housing price records from 2013 to 2019 for the regression task to predict property prices based on feature values. We treat each calendar year as a distinct domain, using 2013–2018 data for training and the final (2019) domain for testing. Concept drift emerges naturally from temporal economic shifts and market fluctuations over the years.

**Appliances Energy Prediction**[6]**.** This dataset addresses regression modeling for predicting appliance energy consumption in a low-energy building. Comprising 10-minute sensor readings over 4.5 months in 2016, it is partitioned into 9 chronological domains. We train on the first eight domains and evaluate on the final (most recent) ninth domain, with concept drift arising from temporal shifts in energy usage patterns across the observation period.

## C.2  Baselines

**Time-agnostic methods.** These methods do not consider the temporal drift, including Offline train on all source domains, train on the last source domain (LastDomain) and incrementally train on each source domain (IncFinetune).

**Continuous Domain Adaptation (CDA).** CDOT [33] predicts the future by transporting labelled samples of the last observed domain to an estimated target distribution via optimal transport, then retrains the classifier on those transported points. CIDA [22] leverages adversarial alignment with a probabilistic domain discriminator to model continuous domain shifts, while its probabilistic extension (PCIDA) enforces higher-order moment matching via Gaussian parameter prediction.

**Temporal Domain Generalization (TDG).** GI [6] regularizes temporal complexity by supervising the first-order Taylor expansion of a time-sensitive model, enabling smooth adaptation to distribution shifts via adversarial selection of temporal perturbations. LSSAE [34] addresses evolving domain generalization by employing a latent structure-aware sequential autoencoder to model dynamic shifts in both data sample space (covariate shift) and category space (concept shift). It leverages variational inference and temporal smoothness constraints to capture continuous domain drift, enhancing generalization to unseen target domains. DDA [9] leverages an attention-based domain transformer to capture directional domain shifts and simulate future unseen domains through bi-level optimization with meta-learning. DRAIN [8] assumes that model parameters vary over time within a fixed network architecture and employs a recurrent neural network to autoregressively predict domain-optimal parameters for future timesteps through temporal dependency modeling.

**Continuous Temporal Domain Generalization (CTDG).** EvoS [35] proposes a multi-scale attention module (MSAM) to model evolving feature distribution patterns across sequential domains, dynamically standardizing features using predicted statistics to mitigate distribution shifts while employing adversarial training to maintain a shared feature space and prevent catastrophic forgetting. Koodos [12] models data and model dynamics as a continuous-time system via Koopman operator theory. It integrates prior knowledge and multi-objective optimization to synchronize model evolution with data drift.

All baseline results presented in Table 1 were directly sourced from their respective original publications to ensure accurate and fair comparison.

## C.3  Implementation Details

The architecture and implementation of backbones and prediction models for all datasets align with DRAIN [8]. Specially, both the encoders and decoder employ a 4-layer MLP architecture with layer dimensions $[1024, 512, 128, m]$, where $m = 32$ denotes the dimension of the Koopman operator. All experiments were conducted on a server with 187GB of memory, an Intel(R) Xeon(R) Gold 6226R CPU@2.90GHz, and two A100 GPUs.

---

[5]https://www.kaggle.com/datasets/htagholdings/property-sales
[6]https://archive.ics.uci.edu/dataset/374/appliances+energy+prediction

We adopt the Adam optimizer across all datasets, with distinct learning rates for the prediction module $lr_{pre}$, encoder-decoder module $lr_{co}$, and Koopman module $lr_{ko}$. For the 2-Moons dataset, the coder and Koopman learning rates are set to $lr_{co} = 1 \times 10^{-3}$ and prediction learning rate $lr_{pre} = 1 \times 10^{-2}$, regulated by $\tau = 0.9$, $\alpha = 10$, $\beta = \gamma = 1$. The Rot-MNIST configuration retains $lr_{pre} = lr_{co} = lr_{ko} = 1 \times 10^{-3}$ and $\tau = 0.9$, $\alpha = 0.1$, $\beta = \gamma = 1$. For ONP, we use $lr_{co} = 1 \times 10^{-2}$ for coder/Koopman and $lr_{pre} = 1 \times 10^{-3}$ for prediction, combined with $\tau = 0.8$, $\alpha = 0.1$, $\beta = 1$, $\gamma = 0.01$. The Shuttle dataset employs a uniform learning rate $1 \times 10^{-3}$ for all modules, and $\tau = 0.9$, $\alpha = \beta = \gamma = 1$. For Elec2, $lr_{pre} = 1 \times 10^{-2}$, $lr_{co} = 1 \times 10^{-4}$ and $lr_{ko} = 1 \times 10^{-3}$ governed by $\tau = 0.1$, $\alpha = 10$, $\beta = 0.1$, $\gamma = 1$. The House dataset shares the coder/Koopman learning rate $1 \times 10^{-3}$ with prediction rate $1 \times 10^{-2}$ with $\tau = 0.3$, $\alpha = 0.1$, $\beta = 10$, $\gamma = 1$. Finally, Appliance maintains a uniform learning rate $1 \times 10^{-3}$ across all modules with $\tau = 0.8$, $\alpha = 1$, $\beta = 1$, $\gamma = 100$.

## D  Supplementary Experiments

### D.1  Visualization of Real-world Uncertainties

To illustrate the challenge of domain-specific uncertainties, Figure 5 visualizes t-SNE embeddings of the first domain ($\mathcal{D}_1$) from two real-world datasets, Elec2 and Appliance, with overlaid Kernel Density Estimates (KDE) highlighting distributional structures. On both real-world datasets, the data are not uniformly distributed; instead, they form multiple distinct high-density regions alongside sparser, peripheral points. These peripheral points, identified as Uncertainties/Noise, typically reside at the fringes of core data concentrations. Such observed heterogeneity within a single temporal domain underscores the frequent violation of the IID assumption. The co-existence of these localized concentrations and scattered uncertainties implies that models attempting to uniformly fit all data within a domain risk overfitting to these domain-specific uncertainties, hereby impairing its ability to generalize to subsequent evolving domains. This vulnerability is precisely what FreKoo aims to mitigate through its targeted handling of different spectral components.

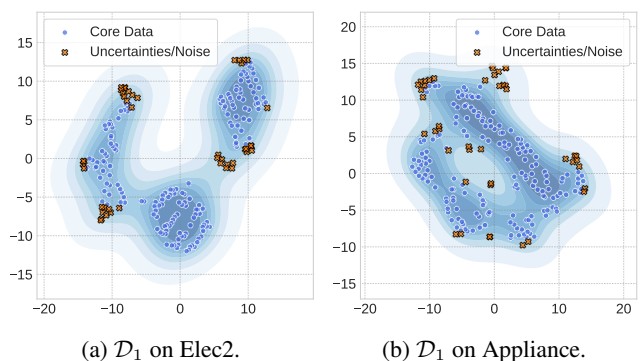

(a) $\mathcal{D}_1$ on Elec2.  (b) $\mathcal{D}_1$ on Appliance.

Figure 5: Visualization of domain-specific uncertainties on real-world datasets.

### D.2  Qualitative Analysis of Decision Boundary

To provide a qualitative assessment of generalization capabilities, we visualize decision boundaries on the 2-Moons target domain, comparing FreKoo against representative TDG (DRAIN [8]) and CTDG (Koodos [12]) methods. As illustrated in Figure 6a, the DRAIN method exhibits a decision boundary that, while attempting to separate the classes, appears somewhat convoluted and potentially overfitted to the specific distribution of the training domains. This can lead to suboptimal generalization on the unseen target domain. Figure 6b shows the boundary learned by Koodos. While demonstrating a degree of adaptation, the boundary still shows some irregularities and does not perfectly capture the underlying structure of the target distribution. In contrast, FreKoo (Figure 6c) learns a notably smoother and more globally consistent boundary that effectively captures the target 2-Moons structure with less sensitivity to local variations. This smoother boundary is indicative of a more robust generalization, suggesting that FreKoo's frequency-domain parameter analysis and dual modeling strategy successfully disentangle stable underlying dynamics from transient fluctuations, leading to a more principled adaptation to the concept drift. This qualitative evidence aligns with the quantitative results, highlighting FreKoo's superior ability to generalize in temporally evolving environments.

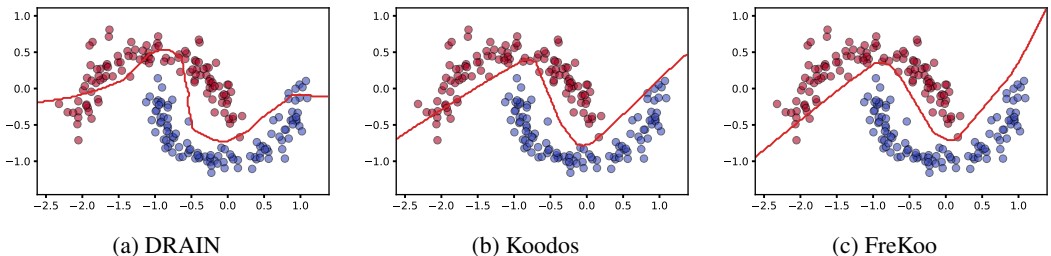

| (a) DRAIN | (b) Koodos | (c) FreKoo |

Figure 6: Visualization of decision boundaries on the 2-Moons dataset. Blue dots and red stars represent different data classes.

### D.3 Qualitative Analysis of Frequency Analysis

Figure 7 visualizes the evolution of model parameters (averaged across dimensions) on P-2-Moons under varying spectral energy preservation ratios $\tau$. FreKoo decomposes the 'Raw Parameter' trajectory into a 'Low-Freq Component,' which captures the smoother, dominant underlying trends, and a 'High-Freq Component,' which encapsulates more rapid, transient fluctuations often indicative of domain-specific noise or artifacts. As shown, the 'Low-Freq Component' (dashed magenta) isolates the core dynamic structure. The 'Reconstructed Parameter' (dashed green), derived from extrapolating the low-frequency dynamics and incorporating regularized high-frequency information, exhibits significantly enhanced smoothness compared to the raw trajectory. This targeted spectral separation and dual modeling strategy allows FreKoo to learn a more stable parameter evolution, mitigating overfitting to transient domain-specific details and thereby fostering improved generalization to future temporal domains. The choice of $\tau$ modulates the trade-off between fidelity to the core dynamics and robustness to high-frequency noise.

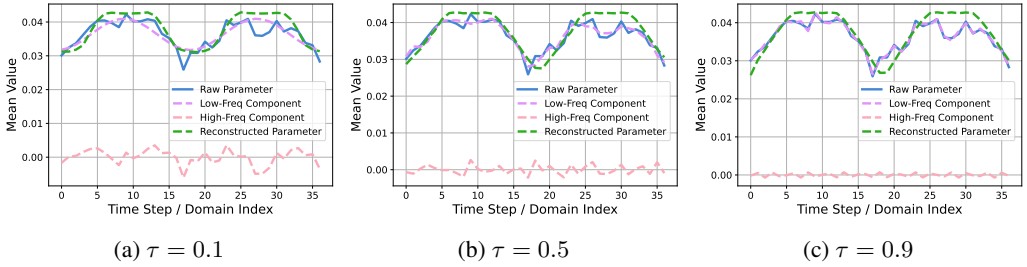

| (a) $\tau = 0.1$ | (b) $\tau = 0.5$ | (c) $\tau = 0.9$ |

Figure 7: Visualization of parameter evolution for different components on the P-2-Moons dataset.

### D.4 Running Time Analysis

We conducted running time tests on three classification datasets and one regression dataset, comparing against DRAIN, the most classic TDG method. We used the same batch size and number of training epochs for both methods to ensure a controlled comparison. As shown in Table 3, FreKoo's training time is competitive

Table 3: Comparison of training time (seconds) between our method and two baselines across two datasets for classification tasks and one datasets for regression tasks.

| Datasets | 2-Moons | ONP | Elec2 | Appliance |
|----------|---------|-------|-------|-----------|
| DRAIN    | 17.28   | 34.45 | 23.30 | 20.23     |
| FreKoo   | 14.26   | 34.83 | 17.19 | 20.42     |

and comparable to that of DRAIN. This demonstrates that the additional components in our framework, such as the spectral decomposition and Koopman operator, do not introduce a significant computational overhead. Crucially, while maintaining similar efficiency, FreKoo consistently outperforms DRAIN in terms of generalization performance. This highlights that our method provides superior performance without sacrificing practical feasibility.

