# OpenReview forum: "Learning Robust Spectral Dynamics  for Temporal Domain Generalization"
_NeurIPS.cc/2025/Conference — NeurIPS 2025 poster_

### Official Review · Reviewer_mg4n · 2025-06-19

**Clarity:** 4
**Significance:** 3
**Originality:** 4
**Rating:** 6
**Confidence:** 5

**Summary:**

This paper introduces FreKoo, a novel framework for Temporal Domain Generalization (TDG) designed to address concept drift, particularly complex drifts involving long-term periodicity and domain-specific uncertainties. The core idea is to analyze the trajectories of model parameters in the frequency domain. The authors provide theoretical grounding for their approach, including stability guarantees for Koopman prediction, a Bayesian justification for high-frequency regularization, and a multiscale generalization bound. Extensive experiments on synthetic and real-world datasets demonstrate FreKoo's superiority over existing SOTA TDG methods.

**Questions:**

See the Strengths and Weaknesses section.

**Ethical Concerns:**

["NO or VERY MINOR ethics concerns only"]

**Final Justification:**

This paper presents a novel and methodologically sound contribution with strong theoretical foundations. The authors have successfully addressed all previously raised concerns, and the comprehensive experimental evaluation on real-world datasets convincingly demonstrates the effectiveness of the proposed approach.

**Limitations:**

Yes

**Paper Formatting Concerns:**

The paper formatting seems good

**Quality:**

3

**Strengths And Weaknesses:**

Strengths:

1. Addresses Critical TDG Limitations: This paper tackles two critical limitations of current TDG methods: (i) the difficulty of modeling long-term periodic dynamics, and (ii) vulnerability to overfitting domain-specific uncertainties. This is particularly relevant for the complex drift scenarios encountered in real-world applications.

2. Novel Methodology: The proposed method, FreKoo, leverages a novel and well-motivated approach by applying spectral analysis to model parameter trajectories for TDG.

3. Rigorous Theoretical Foundation: The paper provides a rigorous theoretical foundation for the proposed method.

4. Comprehensive Experiments: The thorough experimental setup, encompassing a diverse set of synthetic and real-world benchmarks with various drift types, convincingly validates the paper's claims.

5. Clear and Compelling Presentation: The paper is generally well-written and clearly structured, featuring a compelling problem statement and motivation.

Weaknesses and Suggestions for Improvement:

1. The paper mentions using encoders/decoders $\left(\varphi, \varphi^{-1}\right)$ for the Koopman operator. While Appendix C. 3 specifies MLP architectures, a little more detail in the main text about how K is learned (e.g., is it simply a linear layer optimized via L_koop, or are there specific constraints to encourage stability beyond the loss term?) would be helpful.

2. Lemma 1 provides a tighter bound if K's spectral radius ρ(K) < 1. Is this condition actively enforced during training, or is it an assumption for the tighter bound?

3. FreKoo operates on model parameter trajectories. While this is an effective proxy for concept drift, a brief discussion on the assumed relationship between parameter evolution in frequency bands and the actual shifts in P(X,Y) could add further depth. For instance, does a strong periodic component in θ_low necessarily imply a periodic shift in the data distribution itself?

4. Throughout the paper, there appears to be an interchangeable use of "periodical" and "periodic" when referring to time- based cyclical patterns (e.g., "long-term periodical dynamics" vs. "periodic drifts"). For clarity and consistency, it is recommended to uniformly use "periodic" as it is the more standard adjective in this context.

5. Please check and complete the missing author information for reference [11] (currently listed as "--").

---

> ### Author Rebuttal · Authors · 2025-07-31
>
> We sincerely thank the reviewer for your valuable comments, which significantly strengthen our work. We are particularly grateful for your recognition of our contributions' key strengths: technical novelty, theoretical foundation, comprehensive experiments, and presentation quality. Below we address all concerns point-by-point.
>
> ### **W1: Explanation about $\mathbf{K}$**
>
> **Learning $\mathbf{K}$:** In our implementation, $\mathbf{K}$ is learned via a standard linear layer. It is not learned in isolation but is jointly optimized with the encoders and decoder via backpropagation as defined in our overall objective **Eq. (11)**. Thank you your suggestion, and we will refine this part in the revised version.
>
> ### **W2: Clarification of K's spectral radius ρ(K) < 1.**
> We appreciate your suggestion on the learning mechanism for the transformation $\mathbf{K}$. As the reviewer correctly infers, ensuring the stability of $\mathbf{K}$ is crucial for preventing error explosion and achieving a tighter theoretical bound. To address this, we actively enforce a stability constraint during training in our code (please refer our offered code of the model implementation). Therefore, this condition is not merely a theoretical assumption for the tighter bound in Lemma 1; it is a practical constraint actively enforced in our implementation. This ensures that the learned operator is stable, which in turn helps to minimize error accumulation and leads to the more robust long-term forecasting performance demonstrated in our experiments.
>
> ### **W3: Explanation about the relationship between parameter evolution and data distribution**
>
> We thank the reviewer for this excellent question regarding the connection between parameter and data dynamics.
>
> Our framework's end-to-end learning process directly establishes this relationship. As detailed in **Algorithm 1**, the parameter trajectory $\Theta$ is learned jointly by minimizing the task loss across all data domains. This ensures that the learned parameter evolution is an intrinsic reflection of the shifts in the data distribution $P(X,Y)$.
>
> Therefore, a strong periodical component in $\Theta_{low}$ is indeed a direct consequence of a periodical shift in the data itself, as this is the optimal trajectory for minimizing loss on such data. We empirically validate this relationship in our **Periodicity Modeling experiment (Section 4.4)**. The visualizations in **Figure 3(b)** provide clear evidence that our model successfully captures the induced periodical structure in the parameter space.
>
> Thank you for the insightful question. We will emphasize this connection more explicitly in the revised manuscript.
>
> ### **(W4/W5) "Periodical" vs. "Periodic" and Missing reference information [11]:**
> Thank you for your suggestion. We will revise the manuscript to consistently use "periodical" for improved clarity and uniformity, and will thoroughly verify all content—including references—to ensure accuracy.

---

> > ### Comment · Reviewer_mg4n · 2025-08-02
> >
> > All of my concerns have been addressed, and I have raised my score.

---

> > > ### Author Response · Authors · 2025-08-03
> > >
> > > Thank you very much for your valuable suggestions to improve our paper. Your positive feedback is also truly encouraging and greatly motivates our future research. We sincerely appreciate it.

---

### Official Review · Reviewer_2Mhj · 2025-06-29

**Clarity:** 3
**Significance:** 2
**Originality:** 2
**Rating:** 4
**Confidence:** 5

**Summary:**

This paper introduces a novel method for Temporal Domain Generalization (TDG) named FreKoo that addresses performance degradation in machine learning models caused by complex concept drifts over time. By transforming model parameter trajectories into the frequency domain, FreKoo decomposes temporal dynamics into low-frequency components, which are extrapolated using the Koopman operator to capture long-term trends and periodicities, and high-frequency components, which are regularized to suppress transient noise and domain-specific uncertainties. The authors provide theoretical guarantees through a multiscale generalization bound and demonstrate FreKoo’s effectiveness in modeling stable parameter evolution. Experiments across classification and regression benchmarks show FreKoo outperforms existing TDG methods, particularly in scenarios with periodic or noisy distribution shifts.

**Questions:**

- Theorem 1 focus only on the relationship between domain $T$ and domain $T+1$, while the algorithm aggregates information from all previous domains to predict $T+1$. Can you extend the theoretical analysis to capture the influence of the full trajectory $\{1, 2, ..., T\}$ on the prediction of domain $T+1$?
- The algorithm assumes that optimal model parameters evolve smoothly over time, which may not hold in practice, particularly in deep networks with high-dimensional non-convex loss landscapes. Can you provide empirical or theoretical justification for this assumption? For example, visualizing learned parameter trajectories on real-world datasets or quantifying smoothness across multiple runs could help.
- The evaluation is limited to a static setting with fixed training/testing split (e.g., domains $1$ to $T$ for training, $T+1$ for testing). The authors should including more dynamic and realistic settings, such as: Varying the split index between training and test domains, evaluating on extended horizons (e.g., $T+2$, $T+3$) to test robustness under longer-range drift, using real-world high-dimensional datasets beyond the current benchmarks.
- The method critically relies on the choice of spectral energy preservation ratio $\tau$ and the decomposition into low- and high-frequency components, but these design choices are not deeply explored. Could you provide more systematic analysis on how sensitive the performance is to $\tau$, and whether this threshold could be learned or adapted from data? Additionally, how does the model perform when trained with noisy or misaligned frequency partitions?
- The use of Koopman operator assumes that low-frequency dynamics evolve approximately linearly in a latent space, but this may be violated in highly nonlinear or chaotic temporal processes. Can you discuss the expressivity limits of the Koopman formulation? Including empirical failure cases or comparisons to nonlinear forecasting alternatives (e.g., RNN-based parameter predictors) would be helpful.

**Ethical Concerns:**

["NO or VERY MINOR ethics concerns only"]

**Final Justification:**

The authors provide clarification as well as conduct additional experiment on model complexity. However, I still find the experimental design somewhat limited, particularly in terms of the datasets and evaluation settings considered. Additionally, the theoretical analysis does not offer substantial novel insights and is not strongly aligned with the proposed algorithm. That said, I believe this is a quite good paper overall, and the strengths outweigh the weaknesses. Accordingly, I have decided to increase my score.

**Limitations:**

- The theoretical analysis relies on Lipschitz continuity of neural networks and the linear Koopman operator to model low-frequency dynamics. These assumptions are difficult to guarantee in practice, particularly in high-dimensional and highly nonlinear deep learning settings.
- The theory assumes that the future domain $T+1$ depends only on the immediately preceding domain $T$, neglecting long-range temporal dependencies that are common in real-world sequences. This simplification limits the theoretical alignment with the actual algorithm, which uses all past domains.
- The method assumes that ground-truth model parameters evolve smoothly over time, which may not hold in complex real-world scenarios. In practice, multiple optimal models can exist with significantly different parameters, undermining the assumption of smooth parameter trajectories.
- While the work focuses on academic benchmarks, the use of TDG methods in high-stakes domains (e.g., finance, healthcare, surveillance) could introduce risks if models generalize poorly across time or are misled by transient patterns. The authors should reflect on potential risks of over-reliance on extrapolation-based generalization and the consequences of failure under severe or abrupt distribution shifts.

**Quality:**

2

**Strengths And Weaknesses:**

Strengths

- The proposed idea makes sense. The key innovation lies in modeling model parameter dynamics in the frequency domain, which allows principled separation of stable (low-frequency) and noisy (high-frequency) components.
- Code is provided for reproducibility purposes.

Weaknesses:
- The proposed theoretical results rest on strong assumptions, including the Lipschitz continuity of the neural networks and the dependence of the future domain $T+1$ solely on the immediately preceding domain $T$ via the Koopman operator. These assumptions are overly restrictive. For instance, enforcing Lipschitz continuity may not be practical for deep neural networks, which often exhibit highly non-smooth behavior. Additionally, assuming that domain $T+1$ depends only on domain $T$, and is conditionally independent of all earlier domains ($1, 2, \dots, T-1$), oversimplifies the complexity of real-world temporal dynamics, which often exhibit long-range dependencies.
- There is a notable gap between the theoretical analysis and the proposed algorithm. Specifically, Theorem 1 characterizes the relationship between domains $T$ and $T+1$ only, whereas the algorithm leverages information from all preceding domains to estimate domain $T+1$. Developing a theoretical bound that captures the influence of all prior domains [1] would provide a stronger alignment between theory and algorithm, leading to a more coherent and rigorous framework.
- The proposed algorithm assumes the existence of ground-truth or optimal model parameters that evolve smoothly over time. However, this assumption is difficult to satisfy in practice, particularly in high-dimensional settings involving complex models and data. For instance, deep neural networks may admit multiple optimal solutions with significantly different parameter sets. This ambiguity challenges the practicality and applicability of the proposed approach in real-world scenarios.
- The experimental design lacks comprehensiveness. The authors evaluate their method only in a static setting, where the model is trained on all but the last domain and tested solely on the final domain. To more thoroughly assess the effectiveness of the proposed algorithm, additional experimental settings should be considered. For example, a dynamic setting with varying split points between training and testing domains [2] would better reflect real-world scenarios. Evaluating model performance on further future timestamps (e.g., $T+2, T+3$) is also recommended to assess long-term generalization. Furthermore, experiments on real-world, high-dimensional datasets [3,4] are crucial to demonstrate the practical utility and scalability of the method.
- The method is designed for batch TDG and doesn't support continual or online adaptation, limiting its applicability in real-time streaming environments.
- The architecture involves multiple encoders and a Koopman operator, and work on parameter space instead on data space, which may introduce nontrivial computational and memory overhead compared to simpler TDG baselines.

References:

[1] Pham, Thai-Hoang, Xueru Zhang, and Ping Zhang. "Non-stationary domain generalization: theory and algorithm." Uncertainty in artificial intelligence: proceedings of the... conference. Conference on Uncertainty in Artificial Intelligence. Vol. 2024. 2025.

[2] Yao, Huaxiu, et al. "Wild-time: A benchmark of in-the-wild distribution shift over time." Advances in Neural Information Processing Systems 35 (2022): 10309-10324.

[3] Ginosar, Shiry, et al. "A century of portraits: A visual historical record of american high school yearbooks." Proceedings of the IEEE International Conference on Computer Vision Workshops. 2015.

[4] Lin, Zhiqiu, et al. "The clear benchmark: Continual learning on real-world imagery." Thirty-fifth conference on neural information processing systems datasets and benchmarks track (round 2). 2021.

---

> ### Author Rebuttal · Authors · 2025-07-30
>
> We appreciate your insightful feedbacks. To improve clarity, we have reorganized the comments from weaknesses/questions/limitations into a unified structure (order may differ from the original). This helps streamline the review process for reviewers/ACs/SACs.
>
> ### **1. Task and setting**
> **1.1. Static vs. continuous settings (W4/W5/Q3)**: Our experimental design ensures fair comparison with SOTA TDG baselines [6,8,12]—the established standard in this field, i.e., discrete domains. Your mentioned concerns are talked in **Limitations (Lines 376-377)**. Crucially, we considered the limitation arising from this discrete domain segmentation in **Introduction (Lines 45-51)**. This represents a significant advance over conventional TDG approaches. In addition, we acknowledge that we did not consider the continuous TD. However, we have included the continuous TD methods (*EvoS[nips2024]，Koodas[nips2024]*) as the comparision, demonstrating our method's competitive performance. Our **novel spectral analysis framework for parameter trajectory prediction** remains the primary contribution, rigorously validated through theoretical analysis and experiments.
>
> **Abrupt distribution shifts**: As discussed in **Related Works (Lines 119-129)**, TDG fundamentally aims to leverage predictable evolutionary dynamics for proactive generalization. Abrupt shifts usually means unpredictable and thus violate the core assumption of TDG methods including ours, which is beyond this paper's scope.
>
> **1.2. (W4)**: While we did not use your suggested datasets ["A century of portraits: A visual historical record of american high school yearbooks"; "The clear benchmark: Continual learning on real-world imagery"], our evaluation was conducted on **5 real-world benchmarks** which are popular in TDG and concept drift. They feature varying numbers of features, complex drift patterns and real-world uncertainties. The consistent SOTA performance across these diverse tasks demonstrates the practical utility and robustness of FreKoo in real-world scenarios.
>
> We sincerely appreciate your valuable suggestions and references. These will be incorporated in the revised manuscript.
>
> ### **2. Assumptions (W1/W3/Q2)**
> **2.1. Lipschitz assumption (W1)**: The Lipschitz assumption is a widely-used tool in the theoretical analysis of domain adaptation and generalization [*"Continuous temporal domain generalization, nips2024"; "Fine-grained analysis of optimization and generalization for overparameterized two-layer neural networks, ICML2019"*]. **The primary purpose of this assumption is to provide a formal framework for analyzing how errors in the latent parameter space propagate to the final prediction risk**. On the one hand, the high-frequency component is not forward-predicted but regularized via temporal smoothness, which naturally suppresses sharp transitions in the parameter space. This acts as an implicit regularizer that effectively controls local Lipschitz behavior in practice. On the other hand, while the Lipschitz constant of deep neural networks is theoretically unbounded, our implementation employs weight decay. This standard regularization technique penalizes large weights, which in turn empirically constrains the model's complexity and helps control its Lipschitz constant in practice.
>
> Thank you so much for raising this point to help enhance the theoretical soundness of our work. We shall provide further clarification regarding these assumptions in the revised version
>
> **2.2 Smooth parameter evolution (W3/Q2:)**: We acknowledge that multiple optimal solutions may exist if we treat each domain's optimization independently. However, our method's core strength is its joint optimization (Eq.(11)) in an end-to-end way, which selects a solution sequence $[\theta_1, \theta_2, \dots, \theta_T]$ that explicitly enforces a coherent temporal structure. This objective function explicitly regularizes the learned parameter trajectory to exhibit desirable temporal properties. By minimizing this, our framework navigates the complex solution space to select a sequence of parameters that not only performs well on historical tasks but is also structurally predictable and smooth.
>
> ### **3. Long-range dependencies (W2/Q1)**
> **3.1. Our method captures long-range dependencies from $0 \rightarrow T$:**
>  * Spectral decomposition is applied to the entire historical parameter trajectory $\Theta = [\theta\_1, \theta\_2, ..., \theta_T]$. This transformation inherently encodes the full history of parameter evolution into the frequency domain. Long-range dependencies naturally manifest as concentrated energy in the low-frequency components ($\Theta\_{low}$).
> * The Koopman operator learns the dynamics on the whole low-frequency latent representations via Eq.(6). Since $\Theta\_{low}$ already encapsulates information from the full history, the learned evolution $\hat{\theta}_{t+1} = \phi^{-1}(\mathbf{K} \phi\_{low}(\theta\_{t, low}) + \phi\_{high}(\theta\_{t, {high}}))$ is effectively propagating a state that is imbued with historical context.
>
> **3.2 Theorem 1:** The primary goal of this theorem is to offer a qualitative decomposition of the generalization error. It provides a principled justification for our algorithm's dual-component design by showing how the total risk can be broken down into: the low-frequency dynamics ($\mathcal{E}\_{low}$) controlled by the stability of the Koopman operator (Lemma 1), and the high-frequency noise ($\mathcal{E}\_{high}$) managed by temporal smoothing regularization (Lemma 2). While the bound is presented for a single step, the logic applies inductively, and the state used for prediction already contains the information from all prior $T$ domains.
>
> **We also extended it**.    We focus on the propagation of the low-frequency latent error $e_{t,low}= \hat{z}\_{t,low}- z^{\*}\_{t,low}$. By recursively analyzing the single-step error recurrence $e_{t+1,low} = K e_{t,low} + e_{t,low}$, where $e_{t,low} = K z^{\*}\_{t,low} - z^{\*}\_{t+1,low}$ is the single-step model mismatch, we can express the error after an $h$-step predictions as:
>
> $
> e_{t_0+h,low} = K^h e_{t_0,low} + \sum_{i=0}^{h-1} K^i e_{t_0+h-1-i, low}
> $
>
> Taking the norm, we get the multi-step error bound:
>
> $
> \|e_{t_0+h,low}\| \le \|K^h\| \cdot \|e_{t_0,low}\| + \sum_{i=0}^{h-1} \|K^i\| \cdot \|e_{t_0+h-1-i, low}\|
> $
>
> This formulation reveals that the final error is not only dependent on the **initial error** ($e_{t_0,low}$), but also on the **sum of all previous errors** ($e_{t,low}$) accumulated along the entire prediction path.
>
> This provides a stronger justification for our full-trajectory optimization approach: The stability of $K$ **Lemma 1** bounds $\|K^i\|$ and controlls the propagation of intermediate errors; The objective function minimizes losses across all domains, directly works to reduce the magnitude of each single-step mismatch error $\|e_{t,low}\|$, thereby minimizing the second term in the bound. A more detailed discussion will be included in the revised appendix.
>
> ### **4. Computational and memory overhead (W6)**
> The computational cost of the additional components is modest. The **Fourier Transform** is efficient (O(T log T)) and applied only to the short sequence of $T$ domains. The encoders and Koopman operator are lightweight MLPs and a linear layer adding minimal overhead. To prove this, we compared the running time with the most popular DRAIN method.
> |Datasets|2-Moons|ONP|Elec2|Appliance|
> |-|-|-|-|-|
> |DRAIN|17.28|34.45|23.30|20.23|
> |FreKoo|14.26|34.83|17.19|20.42|
>
> As the results show, FreKoo's training time is competitive and comparable to that of DRAIN. We think the introduced overhead is manageable and highly justified by its superior performance. The added components are vaulable for modeling the complex dynamics that simpler methods fail to capture.
>
> ### **5. Analysis of $\tau$ (Q4)**
> We analyzed the sensitivity to the energy preservation ratio $\tau$ in **Section 4.5**. The optimal $\tau$ reflects a principled trade-off: higher $\tau$ is better for clean structured drifts (2-Moons), while a more moderate $\tau$ excels on noisy real-world data (Appliance) by filtering out disruptive high-frequency components. We explicitly highlight adaptive threshold $\tau$ as a future work in our **Limitation (Lines 374-376)**. Our current work establishes the foundational effectiveness of using a fixed partition, paving the way for these more advanced adaptive methods.
>
> ### **6. Discuss about Koopman operator (Q5)**
> We thank the reviewer for this excellent and insightful question.
>
> Rather than assuming linearity in the original system, Koopman theory finds a latent space (via a powerful nonlinear mapping $\phi$ learned by the encoder) that can be linearized [*A data–driven approximation of the koopman operator: Extending dynamic mode decomposition, Journal of Nonlinear Science 2015*]. Theoretically, $\phi$ can approximate many nonlinear systems (per the universal approximation theorem) given sufficient capacity.  In this sense, Koopman theory provides a framework for modeling nonlinear dynamics. However, for fully chaotic systems with irregular or unpredictable dynamics, we believe modeling becomes inherently difficult, and the Koopman framework may also fail. This may go beyond the task assumptions of TDG, but it remains an important and thought-provoking direction for future exploration.
>
> **FreKoo vs. RNN-based methods**: Actually, DRAIN is an RNN-based method. However, DRAIN attempts to fit noisy parameter trajectories directly with a single nonlinear model. In contrast, FreKoo simplifies the learning process by explicitly decomposing the trajectories into low-frequency (signal) and high-frequency (noise) components using the Fourier transform, before modeling the dynamics. Our experimental results (see Table 1) also show that FreKoo consistently outperforms DRAIN.

---

> > ### Comment · Reviewer_2Mhj · 2025-08-04
> >
> > Thank the authors for the clarification and the additional experiment on model complexity. I like the idea of using frequency domain for temporal domain generalization. However, I still find the experimental design somewhat limited, particularly in terms of the datasets and evaluation settings considered. Additionally, the theoretical analysis does not offer substantial novel insights and is not strongly aligned with the proposed algorithm. That said, I believe this is a quite good paper overall, and the strengths outweigh the weaknesses. Accordingly, I have decided to increase my score.

---

> > > ### Author Response · Authors · 2025-08-04
> > >
> > > We sincerely appreciate your valuable suggestions and positive feedback. Your comments have also given us important directions for future research, especially in tackling more complex scenarios and theory innovations. Your recognition is a great encouragement and motivation for our continued efforts. Thank you very much!

---

### Official Review · Reviewer_kzhb · 2025-06-30

**Clarity:** 2
**Significance:** 2
**Originality:** 3
**Rating:** 5
**Confidence:** 4

**Summary:**

In time-series analysis, out-of-distribution generalization remains a significant challenge. The paper tackles the setting where the label-generating mechanism changes over-time. The goal is to be able to predict these changes ahead of time and adapt the model.

The proposed method treats the model parameters themselves as the signal. At every time step, an FFT vector is computed from the parameter trajectory estimated thus far. The learner separates the low-frequency components and the high frequency of the FFT trajectory and learns to predict the trajectory governed by the low frequency components and high frequency components. For the low-frequency part, the model uses the low frequency component $\theta_t^{\text{low}}$ and passes it via an encoder to output a latent. The latent evolution is modeled via a linear transformation (this part of the work is inspired from Koopman operators). For the temporal trajectory of the high-frequency components, a temporal difference regularization is imposed to smoothen out the changes.

The authors develop a multi-scale generalization bound to explain why this philosophy of frequency aware separation helps with improved generalization. Finally, the authors conduct experiments on seven benchmarks out of which five are classification based and two are regression based. In these experiments, the authors both establish that their approach turns out to be state-of-the-art but also provide an ablation analysis to show critical the role of different modeling components (Koopman operator, frequency decomposition etc.).

**Questions:**

Please refer to the weaknesses section to read the questions.

**Ethical Concerns:**

["NO or VERY MINOR ethics concerns only"]

**Final Justification:**

The authors have succesfully addressed my key concerns regarding writing and usage of Koopman operator.

**Limitations:**

While the authors have a limitations section, the discussion of the limitations is rather sparse and can be expanded.

**Paper Formatting Concerns:**

I did not notice any formatting concerns.

**Quality:**

3

**Strengths And Weaknesses:**

**Strengths**

1. **Technical novelty** Treating the parameters as signal and separation of the signal into low and high-frequency component is a new method that to the best of my knowledge was not tried before.

2. **Theoretical arguments** Low-frequency component's evolution depends on the Koopman operator. The authors show the sharpening of the generalization bound provided the spectral radius of the operator is less than one. Further, the authors propose a Bayesian justification for the smoothening regularizer used for the high-frequency components.

3. **Solid empirical section** The results in the experiments section show that the approach is especially beneficial in modeling periodic drifts as shown by Elec 2 and Appliance dataset. The ablations and sensitivity plots are quite informative.

**Weaknesses**

1. **Writing** I found the paper poorly written. The method is hard to follow as it is broken down into many small subparts. The authors made the unfortunate choice of having the algorithm in the Appendix. The authors should take the pain of writing a simple step-by-step description of the key elements of the method. Especially since the method would first involve learning theta's for each domain, it makes it all the more confusing as that is not done in standard methods.

2. **Vague claims:**  First let me quote, "Prevailing TDG methods often fall short under complex concept drifts. Typically constrained by assumptions of incremental change or local smoothness, they struggle to simultaneously capture long-range dynamics like periodicity (Challenge 1) and filter transient noise or domain-specific uncertainties (Challenge 2), thereby failing to effectively balance stability with adaptability crucial for real-world scenarios"
Firstly, both these challenges are vaguely stated. Can the authors be more precise on what do each of these challenges mean in a mathematical sense?
Secondly, when the authors state that the existing TDG methods fail to capture xyz, it would be best if authors can create a simple toy example. Through this toy task, the authors can highlight why the existing methods fail and why their method succeeds. Bear in mind, the toy task is to build intuition.
Let me quote "This averaging operation inherently suppresses parameter-wise variations in oscillation patterns, ensuring robustness to localized noise in individual dimensions". Now this is again a vague claim, which is not precise and has no evidence to back it up. Can the authors explain the choice of averaging better? or was it just the most natural heuristic?


3. **On the role of Koopman operator** In the current form, the authors use the Koopman operators to motivate their loss function. I found the whole description via Koopman operators overloaded and unnecessary. The authors might have as well stated that they are modeling latent dynamics via a linear transformation. Here is a parallel I want to draw. Consider kernel SVMs, where one motivates featurization in infinite dimensional spaces and then uses representer theorems to show how Kernel matrix is all we need to characterize the solutions. In those descriptions, the infinite dimensional formulation is actually used. In this case, the authors are referring to an infinite dimensional object but using the straightforward linear usage of it. It is like referring to kernel SVMs, but invoking the linear kernel. Hence, I recommend that the authors tone down the jargon as it feels unncessary.


4. **On the method** Overall I find that the method currently involves several heuristic choices and is not elegant. And one important lesson that stands test of time, is that simpler methods seem to outshine more complex counterparts in the long run. If in spirit of this, the authors could think of simplifications that they can do and suggest some simplifications for future work, that would be nice.
Did the authors consider a simple transformer-based baseline? The transformer can take in as input the parameters theta and predict their changes? It will be nice to illustrate the failures of a transformer based baseline to model these long-term periodic changes in the parameters.

---

> ### Author Rebuttal · Authors · 2025-07-30
>
> We sincerely thank the reviewer for your valuable feedbacks. We are also grateful for the reviewer's recognition of our work's strengths, including its technical novelty, theoretical arguments, and solid empirical results. We address your concerns and questions below.
>
> ### **W1：Writing**
>
> **1. Writting logic:** We apologize for making confusion, and we briefly outline the logical flow to hopefully provide a clearer step-by-step picture:
>
> *   **Step 1:** A crucial point we wish to clarify is that our method is an **end-to-end joint optimization process**. We do not first discretely learn an independent optimal $\theta_t$ for each domain $\mathcal{D}_t$. Instead, the entire parameter trajectory $\Theta = [\theta_1, \theta_2, ..., \theta_T]$ is learned jointly by minimizing our overall objective function (**Eq. 11**). We will claim this in the revised verson.
>
> *   **Step 2: Spectral Decomposition (Section 3.2.1)**: Within this end-to-end learning loop, we apply the Discrete Fourier Transform (DFT) to decompose parameter trajectory into a **low-frequency component $\Theta_{low}$** (capturing stable, long-term trends and periodicity) and a **high-frequency component $\Theta_{high}$** (representing transient noise and uncertainties). This decomposition is the core of our frequency-aware perspective.
>
> *   **Step 3: Dual-Strategy Dynamics Modeling (Section 3.2.2)**: We then apply a distinct strategy to each component: For $\Theta_{low}$, we learn a **Koopman operator $\mathbf{K}$** to model its dominant dynamics in a latent space. This is guided by the $\mathcal{L}\_{{rec}}$ loss. For $\Theta_{high}$, we apply a **temporal smoothing regularization ($\mathcal{R}\_{{high}}$)**. This prevents overfitting to domain-specific noise/uncertainties.
>
> *   **Step 4:** Finally, we give the prediction for the future parameter $\hat{\theta}_{t+1}$ and the joint optimzation objective, following by the Theoretical analtsis in Section 3.3
>
> **2. Position of Algorithm 1:** We are sorry to place Algorithm 1 in the Appendix due to the strict page limitations for the initial submission. Thank you for pointing out its importance.
>
> We are grateful for your suggestions, and we will carefully revise the manuscript to improve its clarity and flow.
>
> ### **W2：Vague claims**
> **1. Explanation of challenges and current limitations.** We apologize for the lack of mathematical formulations. To clarify:
>
> - **Challenge 1 (Periodicity)**: we refer **recurring drift** defined in [Learning under concept drift: A review. TKDE 2019] to formulate this challenge, i.e.,
> $\exists L \in \mathbb{Z}^{+} :
> P_t(\mathbf{X},Y) \neq P_{t+1}(\mathbf{X},Y)$ and
> $P_t(\mathbf{X},Y) \approx P_{t+kL}(\mathbf{X},Y)
> \forall k \in \mathbb{Z}^{+}$, where $L$ is Period length. Current TDG methods assume monotonic/smooth drift [6,8,12] failing to model such periodicity, and **Section 4.4 experiment** provide a experimental evidnece.
>
> - **Challenge 2 (Uncertainties):** Characterized by **transient noise and domain variations** that violate IID assumptions within temporal segments (Intro L45-48). While mathematically challenging to formalize, we provide intuitive visual evidence of these uncertainties in real-world scenarios through **Appendix Figure 5**. To our knowledge, no existing TDG method explicitly addresses this challenge.
>
> We appreciate your suggestion again and will refine these points in the revision.
>
> **2:** We apologize for making confusion regarding the **averaging operation for computing spectral energy (Eq.3)**. If we were to select dominant frequencies based on each parameter dimension $d$ individually, we might select different sets of frequencies for different dimensions. This could lead to a fragmented and incoherent model of the overall dynamics. More critically, a single parameter dimension might exhibit a spurious, high-energy oscillation due to optimization noise or artifacts specific to that dimension. A model that latches onto this "localized noise" would not generalize well. By averaging the spectral magnitudes across all $D$ dimensions to get $M_f$, we are effectively performing a pooling operation. This computes a summary statistic that represents the overall importance of frequencyto the evolution of the entire parameter vector. We will revise the manuscript to incorporate these clarifications.
>
> ### **W3: The role of Koopman operator.**
> We sincerely thank the reviewer for this insightful comment and the thoughtful parallel drawn to Kernel SVMs. We would like to clarify that the Koopman framework is not just "jargon," but a **fundamental concept that guides our entire model design**, distinguishing it from a generic linear transformation. The core of our data-driven Koopman approach is to simultaneously learn both the nonlinear feature map $\phi$ (the encoder) and the linear transfermation $\mathbf{K}$. We aim to discover a nonlinear coordinate system via our encoder $\phi$ where the dynamics become globally linear. This is a significant departure from using a fixed feature map, which would be analogous to a "linear kernel". Our deep encoder learns a highly complex, data-driven feature map—a practice established in modern Koopman analysis [Deep learning for universal linear embeddings of nonlinear dynamics. *Nature communications 2018*]. Therefore, a more fitting analogy for our method would be learning the kernel function itself, rather than simply invoking a linear kernel. The "infinite-dimensional" nature of the Koopman operator is addressed by leveraging the universal approximation capabilities of a deep neural network $\phi$ to find an optimal finite-dimensional basis where a linear operator $\mathcal{K}$ suffices. In summary, the Koopman framework is not an unnecessary descriptor but the foundational blueprint for our model's architecture: it justifies the search for a nonlinear embedding $\phi$, dictates the linear structure of the latent dynamics $\mathcal{K}$, and provides the rationale for why this combination is effective for complex temporal dynamics.
>
> We are grateful for this constructive feedback, as it will undoubtedly help us improve the accessibility and impact of our paper. We believe this revised presentation will strike a better balance, making the method easy to follow while still retaining the valuable theoretical context that Koopman theory provides.
>
> ### **W4: On the method**
> We thank the reviewer for this insightful comment on the elegance of our method and for raising the important point about the long-term value of simplicity. We appreciate the opportunity to discuss the design philosophy of FreKoo and the alternatives like Transformers.
>
> **On the method's principled design:**
> While FreKoo involves several components, we think that this is not an arbitrary integration but rather a **principled and structured design** derived from a new **frequency-aware perspective** on TDG. We decompose the parameter trajectory into a stable **low-frequency** component (long-range dynamics) and a transient **high-frequency** component (uncertainties). Each part of our method is a dedicated tool for this decomposition: the Fourier Transform for disentanglement, the Koopman operator for stable extrapolation, and targeted regularization for noise suppression. This structured approach is supported by a rigorous **theoretical analysis (Theorem 1)** and, importantly, provides a novel interpretable solution to TDG by allowing for the analysis of distinct temporal drift components.
>
> **On a Transformer-based Alternative:**
> Thank you for the excellent suggestion of a Transformer-based baseline. A Transformer is indeed a powerful sequence model. We have not yet conducted this comparison, but we believe it is a very valuable direction for future work. Investigating the trade-offs between our structured-bias approach and a powerful general-purpose model like a Transformer would be highly insightful and we are grateful for the suggestion.

---

> > ### Comment · Reviewer_kzhb · 2025-08-03
> >
> > I thank the authors for addressing the concerns. I will raise the score.

---

> > > ### Author Response · Authors · 2025-08-03
> > >
> > > Thank you very much for your insightful comments and constructive suggestions, which really help us think more deeply and broadly about our work. We also sincerely appreciate the improved score. It is a great encouragement and motivates us to further refine our research.

---

### Official Review · Reviewer_MuEU · 2025-07-05

**Clarity:** 3
**Significance:** 3
**Originality:** 3
**Rating:** 5
**Confidence:** 3

**Summary:**

This paper introduces FreKoo for TDG (Temporal Domain Generalization). FreKoo first decomposes the model parameter into low and high frequency parts by DFT. FreKoo models the low frequency part by learnable Koopman operator, and suppresses high frequency part by adding regularizations. This paper gives a generalization bound for FreKoo, and demonstrates its practical advantage through experiments.

**Questions:**

1. Will the number of parameters in the model also affect the performance?
2. Will overparameterization/overfitting of the model affect the selection of high and low frequency components?
3. In appendix C3, the implementation details of the hyperparameters, different datasets and tasks have different learning rate for prediction module, encoder-decoder module, Koopman module. Also, the spectral pereservation ratio and loss weights are different as well. How these hyperparameters were selected? Is there any underlying intuition or guiding principle behind these choices? Furthermore, why do these parameter settings appear to generalize well to all datasets except the appliance dataset?

**Ethical Concerns:**

["NO or VERY MINOR ethics concerns only"]

**Limitations:**

Yes

**Quality:**

3

**Strengths And Weaknesses:**

Strengths:
o The spectral decomposition of the parameter space of the model is interesting and well-grounded.
o Using Koopman operator to model low-frequency components and regularizations for high-frequency components are technically solid.
o The paper provides theoretical results to validate the performance of FreKoo.
o The experiments are extensive, by evaluating across various well-known datasets and benchmarks.
o The paper is well-written and well-organized.

Weaknesses:
o For experiment completeness, please also include the inference and training time. This information is critical for assessing the practical
feasibility of the method, especially in time-sensitive applications like online learning.
o The experimental comparison would be more robust with the inclusion of TKNets, another Koopman-based approach. Incorporating it would provide a clearer understanding of how the proposed method compares against existing alternatives in the same domain.

---

> ### Author Rebuttal · Authors · 2025-07-30
>
> We sincerely appreciate your valuable feedback and are grateful for the reviewer's recognition of our work's technical rigor, theoretical foundation, comprehensive experiments, and clear writing.
>
> ### **W1：Runing time**
>
> We thank the reviewer for this suggestion regarding experimental completeness. We agree that running time analysis is critical for assessing the practical feasibility of our method.
> As shown in the following table, we conducted running time tests on three classification datasets and one regression dataset, comparing against DRAIN, the most classic TDG method. We used the same batch size and number of training epochs for both methods to ensure a controlled comparison. As the results show, FreKoo's training time is competitive and comparable to that of DRAIN. This demonstrates that the additional components in our framework, such as the spectral decomposition and Koopman operator, do not introduce a significant computational overhead. Crucially, while maintaining similar efficiency, FreKoo consistently outperforms DRAIN in terms of generalization accuracy (as shown in Table 1). This highlights the key advantage of our approach: it provides superior performance without sacrificing practical feasibility.
>
> Runing time (s)
> |Datasets|2-Moons|ONP|Elec2|Appliance|
> |-|-|-|-|-|
> |DRAIN|17.28|34.45|23.30|20.23|
> |FreKoo|14.26|34.83|17.19|20.42|
>
> ### **W2：Comparison with TKNets.**
> Thank you for your suggestion. Actually, when we initially designed the experiments, we considered the TKNets method. However, since the datasets used in that paper differ from ours, we had no optimal parameter settings to reference, so we did not include it in our initial comparison. Nevertheless, we have already cited that paper in our work, i.e., Ref[11]. The baseline methods currently compared in our paper were all obtained directly from their original publications, making them more reliable. To address your concern, we have now attempted to re-train TKNet on two classification and one regression datasets. The results are presented in the following table.
> ||2-Moons|Rot-MNIST|Appliance|
> |-|-|-|-|
> |TKNets|3.9 ± 0.4|8.7 ± 0.1|6.6 ± 0.0|
> |FreKoo|1.0 ± 0.3|6.9 ± 0.7|4.0 ± 0.1|
>
> While the TKNets uses of Koopman theory in TDG by modeling the evolution of the data distribution, our FreKoo framework offers a distinct and robust perspective. Instead of analyzing the data space, FreKoo introduces a novel frequency-aware view by directly modeling the dynamics of the model parameters themselves. Our approach is specifically designed to tackle two critical challenges not addressed by prior works: the robust modeling of long-range periodicity and the filtering of domain-specific uncertainties. By decomposing the parameter trajectory into low- and high-frequency components and applying a dual-modeling strategy — Koopman extrapolation for structure and temporal smoothing noise. FreKoo provides a new principled solution that excels in scenarios with complex temporal dynamics. Thank you so much for your suggestion and we will further discuss this point in the revised version.
>
>
> ### **Q1：Will the number of parameters affect the performance?**
> We thank the reviewer for this insightful question. Yes, the number of parameters will affect performance. As with most deep learning models, changes to the architecture—such as modifying the backbone or altering the structure of the encoder-decoder—would lead to a different number of parameters and a corresponding change in performance, which could be either an improvement or a decline. This is a general characteristic of deep models and has also been analyzed in prior works like DRAIN. **To ensure a fair and controlled comparison, we maintained a consistent backbone architecture for all baselines, as detailed in our Appendix C.3**. This approach mitigates the risk of performance differences arising from architectural inconsistencies rather than the novelty of the methods themselves.
>
> Nonetheless, we believe this is an important question. The interplay between model complexity, the number of parameters, and generalization performance in the context of TDG is a valuable area for deeper investigation. We plan to explore this potential relationship in our future work. Thank you again for raising this thoughtful point.
>
> ### **Q2：Will overparameterization/overfitting of the model affect the selection of high and low frequency components?**
> We thank the reviewer for this very insightful question that probes the core of our method. Yes, overparameterization and the resulting overfitting will affect the selection of high and low-frequency components. This is a crucial challenge that we explicitly considered in our design. If a model overfits to the transient noise within each domain $D_t$, the learned parameter trajectory $\Theta$ will become erratic and dominated by high-frequency oscillations. This would "pollute" our spectral decomposition, causing energy to leak from the true low-frequency dynamics into the high-frequency bands, thus corrupting the signal we aim to model. To address this, our FreKoo framework is designed with built-in temporal regularization through its joint optimization objective (**Eq. 11**):
>
> 1.  The **Koopman loss ($\mathcal{L}_{{koop}}$)** penalizes unstructured, unpredictable low-frequency trajectories. This forces the model to find a more regular and stable underlying dynamic, implicitly resisting overfitting.
> 2.  The **high-frequency regularization ($\mathcal{R}_{{high}}$)** directly penalizes sharp, noisy fluctuations in the parameter trajectory.
>
> These components act as a powerful regularizer against overfitting in the temporal dimension. They guide the model to learn a parameter trajectory that is not only accurate on the training data ($\mathcal{L}_{{total}}$) but is also temporally smooth and predictable. This ensures a cleaner and more meaningful separation between the low-frequency signal and high-frequency noise, even in the presence of overparameterization.
>
> ### **Q3：About hyper-parameter Analysis**
> Thank you for your question regarding parameter tuning. For hyperparameters $\alpha$, $\beta$, and $\gamma$, we used traditional grid search over $[0.01, 0.1, 1, 10, 100]$ while fixing other parameters, as described in **Section 4.5**. For the learning rate, we explored $[10^{-4}, 10^{-3}, 10^{-2}]$—a limited range that kept the tuning process manageable. These are standard ML tuning approaches, though admittedly not always adaptive. The spectral energy preservation ratio $\tau$ remains our most important parameter. Different real-world datasets exhibit varying drift types, frequencies, and uncertainties, necessitating dataset-specific tuning. Since $\tau$ is confined to [0,1], this adjustment remains relatively straightforward. Regarding the Appliance dataset, however, our method achieves SOTA (Table 1). Could you please let us know which part is your concern.
>
> Thank you for raising this point. To be honest, parameter tuning remains a less-than-ideal aspect of our approach. We will focus on this area for further research in future work.

---

### Decision · Program_Chairs · 2025-09-17

**Decision:**

Accept (poster)

**Comment:**

This paper introduces a novel method for dealing with temporal distribution shifts, by leveraging Fourier analysis and Koopman operator theory. It addresses a tricky problem in time series analysis by separating low and high frequency components of the signal and using the Koopman operator to extrapolate the former. The paper further comes with a solid experimental evaluation and a theoretical basis.

Some initial concerns about formal clarity and the specific role of the Koopman operator, the model assumptions, training \& inference times, and missing baselines were comprehensively addressed by the authors in their rebuttal through runtime evaluations, additional comparisons, and a number of specific clarifications. Although a few concerns remained about limitations of the experimental setups, all referees uniformly agreed on acceptance, and so this was an easy decision.

A few notes from my side: The focus of this work is on classification and regression tasks, less so on forecasting, which the authors might want to make clearer from the outset. Also, the paper currently lacks a Limitations section, where remaining shortcomings discussed with the referees, like the model’s inability to deal with abrupt distribution shifts could be discussed (the authors argue the latter are essentially unpredictable, but I do not concur with this, see lit. on predicting tipping points).

Overall this is a very good paper and I go with the referees on recommending acceptance.